# NeuCLIP: Efficient Large-Scale CLIP Training with Neural Normalizer Optimization

**Xiyuan Wei**
Texas A&M University
xwei@tamu.edu

**Chih-Jen Lin**
National Taiwan University
cjlin@csie.ntu.edu.tw
Mohamed bin Zayed University
of Artificial Intelligence
chihjen.lin@mbzuai.ac.ae

**Tianbao Yang**
Texas A&M University
tianbao-yang@tamu.edu

## Abstract

Accurately estimating the normalization term (also known as the partition function) in the contrastive loss is a central challenge for training Contrastive Language-Image Pre-training (CLIP) models. Conventional methods rely on large batches for approximation, demanding substantial computational resources. To mitigate this issue, prior works introduced per-sample normalizer estimators, which are updated at each epoch in a blockwise coordinate manner to keep track of updated encoders. However, this scheme incurs optimization error that scales with the ratio of dataset size to batch size, limiting effectiveness for large datasets or small batches. To overcome this limitation, we propose NeuCLIP, a novel and elegant optimization framework based on two key ideas: (i) **reformulating** the contrastive loss for each sample **via convex analysis** into a minimization problem with an auxiliary variable representing its log-normalizer; and (ii) **transforming** the resulting minimization over $n$ auxiliary variables (where $n$ is the dataset size) via **variational analysis** into the minimization over a compact neural network that predicts the log-normalizers. We design an alternating optimization algorithm that jointly trains the CLIP model and the auxiliary network. By employing a tailored architecture and acceleration techniques for the auxiliary network, NeuCLIP achieves more accurate normalizer estimation, leading to improved performance compared with previous methods. Extensive experiments on large-scale CLIP training, spanning datasets from millions to billions of samples, demonstrate that NeuCLIP outperforms previous methods. Code is available at https://github.com/Optimization-AI/NeuCLIP.

## 1 Introduction

Since its introduction, Contrastive Language-Image Pretraining (CLIP) (Radford et al., 2021) has emerged as the de facto standard for vision-language representation learning. The strong capability to align images with corresponding texts has made CLIP valuable for a wide range of real-world applications, including zero-shot classification (Qian & Hu, 2024), cross-modal retrieval (Zeng & Mao, 2022), text-to-image generation (Ramesh et al., 2022), and high-quality dataset selection (Fang et al., 2023a). With the rise of large language models (LLMs), CLIP has also been widely adopted to equip LLMs with the ability to interpret visual inputs (Bai et al., 2025).

A fundamental impediment to training CLIP models is their extensive dependence on huge datasets: attaining competitive performance typically mandates access to millions or even billions of image-text pairs (Fang et al., 2023a; Wang et al., 2025b). A mainstream approach for training CLIP models is to optimize a bimodal contrastive loss, which contrasts each positive image-text pair against numerous negative pairs. To enable training on billions of samples, two primary strategies have emerged to approximate the prohibitive normalization term required for contrastive loss gradient calculation. **The first strategy**, which relies on massive GPU resources, uses an extremely large batch size to construct a contrastive loss within each batch for backpropagation. This strategy is used by many works, including OpenAI CLIP (Radford et al., 2021) and OpenCLIP (Cherti et al., 2023). **The second strategy** addresses the high resource demand of the first by directly optimizing a

global contrastive loss, which contrasts each positive pair with all negative pairs. To tackle the computational challenge, an estimator of the normalizer for each sample's contrastive loss is maintained and updated using a moving average formula following a rigorous framework of finite-sum coupled compositional optimization. It was first proposed by Yuan et al. (2022) for unimodal contrastive self-supervised learning, and later adopted by Wei et al. (2024) with significant improvements for CLIP training, yielding the FastCLIP method. While being less resource demanding, this strategy suffers an inherent limitation that the optimization error scales with the ratio of dataset size to batch size, constraining its effectiveness for large datasets or small batch sizes.

Recently, there emerge some new ideas for CLIP training. For example, Zhai et al. (2023) proposed SigLIP with a sigmoid-based contrastive loss by formulating the problem as a binary classification problem, which avoids the computation of the normalization term involving numerous negative pairs. However, SigLIP still requires a large batch size to achieve competitive performance. Sun et al. (2025) proposed AmorLIP that leverages a lightweight network to predict the normalizer of each contrastive loss. While conceptually appealing, AmorLIP's objective for training the lightweight network still faces the challenge of estimating the log-partition function for the gradient calculation of the lightweight network, leading to a chicken-and-egg problem.

This paper aims to address the limitations of FastCLIP and AmorLIP for CLIP training through a principled approach towards optimizing the global contrastive loss with a neural normalizer. Our method is built on two key ideas: (1) using convex analysis, we reformulate the contrastive loss of each anchor data to a minimization problem with an auxiliary variable, whose optimal solution corresponds to the log-normalizer; and (2) using variational analysis, we transform the minimization over $n$ auxiliary variables (where $n$ is dataset size) into minimization over a compact network that directly predicts the log-normalizers, referred to as the **normalizer-prediction network (NPN)**.

Compared with FastCLIP and AmorLIP, our method offers several notable advantages. First, the objective for learning the encoders and the NPN is unified, and its gradient avoids any nonlinear dependence on the partition function. This allows traditional stochastic gradient methods to be employed for updating both the encoders and the NPN without incurring gradient estimation bias. Second, the unrestricted optimal solution of the auxiliary variable motivates us to inject inductive bias into the design of the NPN, resulting in a simple yet effective architecture: a feedforward layer on top of the encoders followed by a log-sum-exponential pooling layer. Moreover, this seamless optimization framework enables key acceleration techniques, including alternating optimization with multiple NPN updates and periodic re-initialization of the NPN's parameters using sampled updated embeddings, which yields significantly better normalizer approximation. The **contributions** of this paper are threefold:

- We reformulate the contrastive loss into an equivalent form in which the normalization terms are explicitly exposed as optimization variables. This reformulation provides a principled foundation for efficient neural normalizer approximation.

- We introduce a joint optimization problem that learns the encoders and a compact normalizer-prediction network (NPN) with a unified objective, which is derived from variational analysis. We also develop an efficient algorithm for alternatively optimizing the NPN and the CLIP encoders.

- We validate the effectiveness of our approach through extensive experiments on large-scale datasets, showing consistent improvement over existing methods. Comprehensive ablation studies are also conducted to highlight the contribution of different components in our framework.

## 2 RELATED WORKS

**Efficient Training of CLIP Models.** Numerous approaches have been proposed to enhance the efficiency of CLIP model training. Prior works include curating high-quality datasets (Schuhmann et al., 2022; Fang et al., 2023a; Xu et al., 2024; Wang et al., 2024), designing more efficient vision encoder architectures (Fang et al., 2023b; Alabdulmohsin et al., 2023; Chen et al., 2024), applying image-token masking to reduce computational cost (Li et al., 2023b;a), and modifying the geometry of the embedding space (Chou & Alam, 2025; Pal et al., 2025). Additional strategies leverage knowledge distillation to train compact student models (Vasu et al., 2024) or employ a pretrained reference model to steer and accelerate the training of a target model, thereby improving scaling laws (Wei et al., 2025). In contrast, our work is orthogonal to these directions: we focus on improv-

ing the optimization process itself by providing a more efficient and stable method for minimizing the contrastive loss.

**Optimizing the Global Contrastive Loss.** The global contrastive loss was first introduced by Yuan et al. (2022) to address the large batch-size requirement of SimCLR (Chen et al., 2020). They proposed an efficient optimization algorithm, SogCLR, with provable convergence guarantees for unimodal self-supervised contrastive learning. Notably, SogCLR with a batch size of 256 matches the performance of SimCLR with a batch size of 8,192 on ImageNet. Subsequent work by Qiu et al. (2023) provided a distributionally robust optimization (DRO) interpretation of the global contrastive loss, leading to a constrained DRO formulation with individualized temperature optimization. Building on this perspective, Wei et al. (2024) proposed a simplified variant that unifies the temperature parameters into a single scalar. Another line of work explains the global contrastive loss from a probabilistic perspective, showing it as the maximum likelihood estimation of a discriminative model (Wang et al., 2025a). Despite these different viewpoints, all methods rely on optimization techniques similar to SogCLR, which maintain and update per-sample estimators for the normalization term of the contrastive loss. Consequently, they share a key limitation: the optimization error scales with the ratio between the dataset size and the batch size.

**Learning with Auxiliary Networks.** Leveraging auxiliary networks to facilitate training has been widely explored (Shen et al., 2024; He et al., 2020; Su et al., 2025; Kim et al., 2024; Evans et al., 2025; Qiu et al., 2024; Sun et al., 2025). We highlight two closely related works (Qiu et al., 2024; Sun et al., 2025). Qiu et al. (2024) introduced *TempNet*, a network designed to predict personalized temperatures for each sample when training CLIP models with a robust global contrastive loss. Their approach was also motivated by variational analysis that led to the joint optimization of encoders and TempNet. Nevertheless, their optimization algorithm, built on SogCLR, still requires maintaining and updating per-sample estimators of the contrastive loss normalization term, and thus inherits the same limitations of SogCLR. Sun et al. (2025) proposed *AmorLIP*, which optimizes a robust global contrastive loss similar to that of Wei et al. (2024) by jointly learning a lightweight network to approximate the partition function. However, AmorLIP differs from our method in several key aspects: (i) the objective for training the lightweight network is heuristically defined as divergence minimization between its predictions and the true partition function values, which still involves a non-linear function of the normalizer, leading to the chick-and-egg problem; (ii) Amor-LIP simply employs a Multi-Layer Perceptron (MLP) with few layers for the NPN, while our design leverages inductive bias to improve performance, as validated in ablation studies. Furthermore, AmorLIP requires maintaining an exponential moving average (EMA) network of NPN to mitigate the chicken-and-egg issue, since its auxiliary objective still involves estimating a nonlinear function of the normalizer.

## 3 PRELIMINARY

**Notations**. We denote by $\boldsymbol{w}$ the parameters of the CLIP model. Let $\mathcal{S} = \{(\boldsymbol{x}_i, \boldsymbol{z}_i)\}_{i=1}^n$ be a training dataset of $n$ samples, where $\boldsymbol{x}_i$ is an image and $\boldsymbol{z}_i$ is its corresponding text description. The features of image $\boldsymbol{x}_i$ and text $\boldsymbol{z}_j$ output by the CLIP model are denoted by $\boldsymbol{e}_{1,i} = e_1(\boldsymbol{w}; \boldsymbol{x}_i) \in \mathbb{R}^d$ and $\boldsymbol{e}_{2,j} = e_2(\boldsymbol{w}; \boldsymbol{z}_j) \in \mathbb{R}^d$ respectively, where $e_1(\boldsymbol{w}; \cdot)$ and $e_2(\boldsymbol{w}; \cdot)$ denote the image encoder and the text encoder, respectively. The cosine similarity between features $\boldsymbol{e}_{1,i}$ and $\boldsymbol{e}_{2,j}$ is denoted as $s_{i,j} := \cos(\boldsymbol{e}_{1,i}, \boldsymbol{e}_{2,j})$.

The convex conjugate of a function $f : \mathbb{R} \mapsto \mathbb{R}$ is given by $f^*(y) := \max_x y \cdot x - f(x)$. From the Fenchel–Moreau theorem (Rockafellar, 1997, Theorem 12.2) we know that if $f$ is a proper, lower semi-continuous and convex function, the convex conjugate of $f^*$ is equivalent to $f$, i.e., $f(x) = f^{**}(x) := \max_y x \cdot y - f^*(y)$.

**Global Contrastive Loss**. Following Wei et al. (2024), we consider optimizing a robust global contrastive loss defined below:

$$\min_{\boldsymbol{w}, \tau \geq \tau_0} \tau \cdot \frac{1}{|\mathcal{S}|} \sum_{\boldsymbol{x}_i \in \mathcal{S}} \log\left(\varepsilon + g_1(\boldsymbol{w}, \tau; i, \mathcal{S})\right) + \tau \cdot \frac{1}{|\mathcal{S}|} \sum_{\boldsymbol{z}_i \in \mathcal{S}} \log\left(\varepsilon + g_2(\boldsymbol{w}, \tau; i, \mathcal{S})\right) + 2\tau\rho, \quad (1)$$

where

$$g_1(\cdot) := \frac{1}{|\mathcal{S}| - 1} \sum_{\boldsymbol{z}_j \in \mathcal{S}, j \neq i} \exp\left(\frac{s_{i,j} - s_{i,i}}{\tau}\right), \quad g_2(\cdot) := \frac{1}{|\mathcal{S}| - 1} \sum_{\boldsymbol{x}_j \in \mathcal{S}, j \neq i} \exp\left(\frac{s_{j,i} - s_{i,i}}{\tau}\right),$$

$\tau$ is the temperature parameter, $\tau_0, \varepsilon$ are small constants, and $\rho > 0$ is a hyperparameter. In order to solve this problem, we need to compute an estimator of the gradient. In particular, the gradient w.r.t. $\boldsymbol{w}$ is given by

$$\tau \cdot \frac{1}{|\mathcal{S}|} \sum_{\boldsymbol{x}_i \in \mathcal{S}} \frac{1}{\varepsilon + g_1(\boldsymbol{w}, \tau; i, \mathcal{S})} \cdot \nabla g_1(\boldsymbol{w}, \tau; i, \mathcal{S}) + \tau \cdot \frac{1}{|\mathcal{S}|} \sum_{\boldsymbol{z}_i \in \mathcal{S}} \frac{1}{\varepsilon + g_2(\boldsymbol{w}, \tau; i, \mathcal{S})} \cdot \nabla g_2(\boldsymbol{w}, \tau; i, \mathcal{S}). \tag{2}$$

We can see that the terms $\varepsilon + g_1(\boldsymbol{w}, \tau; i, \mathcal{S})$ and $\varepsilon + g_2(\boldsymbol{w}, \tau; i, \mathcal{S})$ serve as **normalizers** of the gradient calculation for image $\boldsymbol{x}_i$ and text $\boldsymbol{z}_i$. A key challenge is that $g_1(\boldsymbol{w}, \tau; i, \mathcal{S})$ and $g_2(\boldsymbol{w}, \tau; i, \mathcal{S})$ cannot be computed exactly, as they depend on all other data samples. This necessitates approximating these normalizers using only a batch of samples.

**Mini-batch Approximation.** A naive approach is to simply use a mini-batch approximation (Radford et al., 2021; Cherti et al., 2023), i.e., sampling a subset $\mathcal{B} \subset \mathcal{S}$ and computing the following gradient estimator

$$\tau \cdot \frac{1}{|\mathcal{B}|} \sum_{\boldsymbol{x}_i \in \mathcal{B}} \frac{1}{\varepsilon + g_1(\boldsymbol{w}, \tau; i, \mathcal{B})} \cdot \nabla g_1(\boldsymbol{w}, \tau; i, \mathcal{B}) + \tau \cdot \frac{1}{|\mathcal{B}|} \sum_{\boldsymbol{z}_i \in \mathcal{B}} \frac{1}{\varepsilon + g_2(\boldsymbol{w}, \tau; i, \mathcal{B})} \cdot \nabla g_2(\boldsymbol{w}, \tau; i, \mathcal{B}).$$

This is equivalent to performing backpropagation on a mini-batch contrastive loss

$$\tau \cdot \frac{1}{|\mathcal{B}|} \sum_{\boldsymbol{x}_i \in \mathcal{B}} \log(\varepsilon + g_1(\boldsymbol{w}, \tau; i, \mathcal{B})) + \tau \cdot \frac{1}{|\mathcal{B}|} \sum_{\boldsymbol{z}_i \in \mathcal{B}} \log(\varepsilon + g_2(\boldsymbol{w}, \tau; i, \mathcal{B})).$$

However, this gradient estimator is biased as its expectation does not give the true gradient in Equation (2) due to the non-linearity of the reciprocal function. As a consequence, it requires a large batch size (Yuan et al., 2022).

**Moving-average Approximation.** To address the large batch issue, Yuan et al. (2022) proposed the SogCLR algorithm, which maintains two sequences of estimators $\{u_{1,i}^{(t)}, u_{2,i}^{(t)}\}$ for each $\boldsymbol{x}_i, \boldsymbol{z}_i$ to approximate $\varepsilon + g_1(\boldsymbol{w}^{(t)}, \tau^{(t)}; i, \mathcal{S})$ and $\varepsilon + g_2(\boldsymbol{w}^{(t)}, \tau^{(t)}; i, \mathcal{S})$ at the $t$-th iteration, respectively. At $t$-th iteration with solutions $(\boldsymbol{w}^{(t)}, \tau^{(t)})$, for $(\boldsymbol{x}_i, \boldsymbol{z}_i)$ in the sampled batch $\mathcal{B}^{(t)}$, their normalizer estimators are updated as follows

$$\begin{aligned} u_{1,i}^{(t+1)} &= (1 - \gamma) u_{1,i}^{(t)} + \gamma(\varepsilon + g_1(\boldsymbol{w}^{(t)}, \tau^{(t)}; i, \mathcal{B}^{(t)})), \\ u_{2,i}^{(t+1)} &= (1 - \gamma) u_{2,i}^{(t)} + \gamma(\varepsilon + g_2(\boldsymbol{w}^{(t)}, \tau^{(t)}; i, \mathcal{B}^{(t)})), \end{aligned} \tag{3}$$

where $\gamma \in [0, 1]$ is a hyperparameter. Then, the gradient estimator for $\boldsymbol{w}^{(t)}$ is computed by

$$\frac{\tau^{(t)}}{|\mathcal{B}^{(t)}|} \sum_{\boldsymbol{x}_i \in \mathcal{B}^{(t)}} \frac{1}{u_{1,i}^{(t+1)}} \cdot \nabla_{\boldsymbol{w}} g_1(\boldsymbol{w}^{(t)}, \tau^{(t)}; i, \mathcal{B}^{(t)}) + \frac{\tau^{(t)}}{|\mathcal{B}^{(t)}|} \sum_{\boldsymbol{z}_i \in \mathcal{B}^{(t)}} \frac{1}{u_{2,i}^{(t+1)}} \cdot \nabla_{\boldsymbol{w}} g_2(\boldsymbol{w}^{(t)}, \tau^{(t)}; i, \mathcal{B}^{(t)}). \tag{4}$$

It has been shown that the optimization error of SogCLR converges to zero (Yuan et al., 2022). Built on this idea, Wei et al. (2024) proposed FastCLIP, an efficient distributed CLIP training framework with several improvements including the temperature optimization and the learning rate schedule for $\gamma$. However, the convergence error of FastCLIP suffers from a scaling factor of $O(n/B)$ on the standard rate (Yuan et al., 2022), where $B = |\mathcal{B}^{(t)}|$ is the mini-batch size per-iteration. This property is not desirable since the error will increase when $n$ increases and $B$ decreases.

# 4 NEUCLIP: CLIP TRAINING WITH NEURAL NORMALIZER OPTIMIZATION

In this section, we first present a reformulation of the contractive loss as a minimization problem. Then we derive a joint optimization problem from variational analysis to learn the encoders and the normalizer-prediction network (NPN). Finally, we present an optimization algorithm.

## 4.1 REFORMULATING THE CONTRASTIVE LOSS

Without lose of generality, let us consider the individual contrastive loss for an image anchor data $\boldsymbol{x}_i$, as given by $F(\boldsymbol{w}, \tau; \boldsymbol{x_i}) = \log(\varepsilon + g_1(\boldsymbol{w}, \tau; i, \mathcal{S}))$. Since $f(\cdot) = -\log(\cdot)$ is a convex function, we

can leverage the conjugate transformation $f(x) = \max_y y \cdot x - f^*(y)$ with $f^*(y) = -\log(-y) - 1$ to reformulate the above individual contrastive loss as follows (by setting $x = \varepsilon + g_1(\boldsymbol{w}, \tau; i, \mathcal{S})$):

$$F(\boldsymbol{w}, \tau; \boldsymbol{x_i}) = \log(\varepsilon + g_1(\boldsymbol{w}, \tau; i, \mathcal{S})) = -\max_y \{y \cdot (\varepsilon + g_1(\boldsymbol{w}, \tau; i, \mathcal{S})) - f^*(y)\}$$

$$= -\max_y \{y \cdot (\varepsilon + g_1(\boldsymbol{w}, \tau; i, \mathcal{S})) + \log(-y) + 1\} = \min_y \{-y \cdot (\varepsilon + g_1(\boldsymbol{w}, \tau; i, \mathcal{S})) - \log(-y) - 1\}$$

$$= \min_\alpha \{\exp(-\alpha) \cdot (\varepsilon + g_1(\boldsymbol{w}, \tau; i, \mathcal{S})) + \alpha - 1\}, \tag{5}$$

where the last equality uses a change of variable $\alpha = -\log(-y)$. It is not difficult to derive that the optimal solution $\alpha^*$ to the last optimization problem is given by $\alpha^* = \log(\varepsilon + g_1(\boldsymbol{w}, \tau; i, \mathcal{S}))$ (cf. Appendix A.1), which is exactly the log-normalizer. We note that the above reformulation can be viewed as a special case of the optimized certainty equivalent (OCE) (Ben-Tal & Teboulle, 2007).

Substituting the above reformulation of each contrastive loss in Equation (1), we get the following equivalent form of the global contrastive loss:

$$\min_{\boldsymbol{w}, \tau} \tau \cdot \frac{1}{|\mathcal{S}|} \sum_{\boldsymbol{x_i} \in \mathcal{S}} \left\{ \min_{\alpha_{1,i}} \exp(-\alpha_{1,i}) \cdot (\varepsilon + g_1(\boldsymbol{w}, \tau; i, \mathcal{S})) + \alpha_{1,i} - 1 \right\}$$

$$+ \tau \cdot \frac{1}{|\mathcal{S}|} \sum_{\boldsymbol{z_i} \in \mathcal{S}} \left\{ \min_{\alpha_{2,i}} \exp(-\alpha_{2,i}) \cdot (\varepsilon + g_2(\boldsymbol{w}, \tau; i, \mathcal{S})) + \alpha_{2,i} - 1 \right\} + 2\tau\rho. \tag{6}$$

Indeed, the update of $u_1, u_2$ sequences of SogCLR in Equation (3) can be recovered by solving the above problem using stochastic block mirror descent method (Lan, 2020, Section 4.6.2). We provide a detailed derivation in Appendix A.2.

## 4.2 Neural Normalizer Optimization

Maintaining and updating $\{\alpha_{1,i}, \alpha_{2,i}\}$ in a coordinate-wise manner is the root that leads to a scaling factor of $O(n/B)$ in the convergence error of FastCLIP. To mitigate this issue, our idea is grounded in the following theorem from variational analysis.

**Theorem 1** (Rockafellar & Wets, 2009, Theorem 14.60). *Let $\mathcal{F}$ be a space of measurable functions from $\Omega$ to $\mathbb{R}$ that is decomposable relative to a finite measure $\mu$. Let $f : \Omega \times \mathbb{R} \to \mathbb{R}$ be a normal integrand. Then, as long as $\int_{x \in \Omega} f(x, \alpha(x))\mu(dx) \neq \infty$ for all $\alpha(\cdot) \in \mathcal{F}$, we have*

$$\inf_{\alpha(\cdot) \in \mathcal{F}} \int_{x \in \Omega} f(x, \alpha(x))\mu(dx) = \int_{x \in \Omega} \left( \inf_{\alpha \in \mathbb{R}} f(x, \alpha) \right) \mu(dx). \tag{7}$$

*Moreover, if the above infimum is not $-\infty$, then $\alpha^*(\cdot) \in \arg\min_{\alpha(\cdot) \in \mathcal{F}} \int_{x \in \Omega} f(x, \alpha(x))\mu(dx)$ if and only if $\alpha^*(x) \in \arg\min_{\alpha \in \mathbb{R}} f(x, \alpha)$ for $\mu$-almost every $x \in \Omega$.*

The above equality indicates that the minimization over individual variables $\alpha$ for each $x$ on the right hand side within an integral of $x$ can be translated into searching for a function $\alpha(\cdot) \in \mathcal{F}$ that minimizes the whole integral over all $x \in \Omega$.

Our reformulated contrastive loss (Equation 6) shares a similar structure to the right hand side of Equation (7) when the measure $\mu$ is a probability measure, with minimization over $\alpha_{1,i}, \alpha_{2,i}$, and then the average over all samples. Hence, Theorem 1 implies that

$$\tau \cdot \frac{1}{|\mathcal{S}|} \sum_{\boldsymbol{x_i} \in \mathcal{S}} \left\{ \min_{\alpha_{1,i}} \exp(-\alpha_{1,i}) \cdot (\varepsilon + g_1(\boldsymbol{w}, \tau; i, \mathcal{S})) + \alpha_{1,i} - 1 \right\}$$

$$= \min_{\alpha_1(\cdot) \in \mathcal{F}} \tau \cdot \frac{1}{|\mathcal{S}|} \sum_{\boldsymbol{x_i} \in \mathcal{S}} \left\{ \exp(-\alpha_1(\boldsymbol{x_i})) \cdot (\varepsilon + g_1(\boldsymbol{w}, \tau; i, \mathcal{S})) + \alpha_1(\boldsymbol{x_i}) - 1 \right\}.$$

As a result, Equation (6) can be transformed into:

$$\min_{\boldsymbol{w}, \tau} \min_{\alpha_1(\cdot), \alpha_2(\cdot) \in \mathcal{F}} \tau \cdot \frac{1}{|\mathcal{S}|} \sum_{\boldsymbol{x_i} \in \mathcal{S}} \left\{ \exp(-\alpha_1(\boldsymbol{x_i})) \cdot (\varepsilon + g_1(\boldsymbol{w}, \tau; i, \mathcal{S})) + \alpha_1(\boldsymbol{x_i}) - 1 \right\} \tag{8}$$

$$+ \tau \cdot \frac{1}{|\mathcal{S}|} \sum_{\boldsymbol{z_i} \in \mathcal{S}} \left\{ \exp(-\alpha_2(\boldsymbol{z_i})) \cdot (\varepsilon + g_2(\boldsymbol{w}, \tau; i, \mathcal{S})) + \alpha_2(\boldsymbol{z_i}) - 1 \right\} + 2\rho\tau.$$

**Neural Normalizer Optimization.** Directly solving (8) is not easier than solving (6) due to the constraint $\alpha_1(\cdot), \alpha_2(\cdot) \in \mathcal{F}$. Our strategy is to approximate these functions using parameterized neural networks. Specifically, we solve the problem by restricting $\alpha_1(\cdot) \in \mathcal{F}_{\boldsymbol{W}_1}$ and $\alpha_2(\cdot) \in \mathcal{F}_{\boldsymbol{W}_2}$, where $\mathcal{F}_{\boldsymbol{W}_1}$ and $\mathcal{F}_{\boldsymbol{W}_2}$ denote the function classes represented by neural networks parameterized by $\boldsymbol{W}_1$ and $\boldsymbol{W}_2$, respectively. This raises the question of how to design the architecture of the neural network. A naive idea is to use simple feedforward neural networks. It is guaranteed by universal approximation theory that a neural network can approximate any continuous function arbitrarily well as long as the network is wide enough. However, this could increase the burden of training. Instead, we draw insights from Theorem 1 to design a network with inductive bias, i.e., by leveraging the structure of the problem. Specifically, we want

$$\alpha_1(\boldsymbol{x}_i) \in \underset{\alpha_{1,i}}{\arg\min} \ \exp(-\alpha_{1,i}) \cdot (\varepsilon + g_1(\boldsymbol{w}, \tau; i, \mathcal{S})) + \alpha_{1,i} - 1$$

$$= \log\left(\varepsilon + \frac{1}{|\mathcal{S}|-1} \sum_{\boldsymbol{z}_j \in \mathcal{S}, j \neq i} \exp\left(\frac{\boldsymbol{e}_{1,i}^T \boldsymbol{e}_{2,j} - \boldsymbol{e}_{1,i}^T \boldsymbol{e}_{2,i}}{\tau}\right)\right), \quad (9)$$

where the last equality is from the optimality condition (cf. Appendix A.1) and the definition of $g_1$. Since $\boldsymbol{e}_{1,i}, \boldsymbol{e}_{2,i}$ are readily available from the CLIP encoders, we only need a model that compresses the information of all $\boldsymbol{e}_{2,j}$. Inspired by this, we define the following network architecture with parameter $\boldsymbol{W}_1 \in \mathbb{R}^{d \times m}$, where $m$ is the number of neurons of the hidden layer:

$$\alpha_1(\boldsymbol{x}_i) := \alpha_1(\boldsymbol{W}_1; \boldsymbol{e}_{1,i}, \boldsymbol{e}_{2,i}) = \log\left(\varepsilon + \frac{1}{m} \sum_{j'=1}^m \exp\left(\frac{\cos(\boldsymbol{e}_{1,i}, \boldsymbol{W}_{1,j'}) - \boldsymbol{e}_{1,i}^T \boldsymbol{e}_{2,i}}{\tau}\right)\right), \quad (10)$$

where $\boldsymbol{W}_{1,j'}$ denotes the $j'$-th column of $\boldsymbol{W}_1$. This can be seen as a compact network built on top of the encoders, processing their output embeddings $\{\boldsymbol{e}_{1,i}, \boldsymbol{e}_{2,i}\}$ with a feedforward layer parameterized by $\boldsymbol{W}_1$ and followed by a log-sum-exponential pooling layer. Compared with the unrestricted optimal solution of $\alpha(\boldsymbol{x}_i)$ in Equation (9), we can view $\boldsymbol{W}_{1,1}, \ldots, \boldsymbol{W}_{1,m}$ as prototypical embeddings that summarize the texts $\{\boldsymbol{z}_j\}$. This is supported by existing studies of self-supervised representation learning, which show that the learned embeddings of training samples from the same class tend to concentrate around their class means (Ben-Shaul et al., 2023). Similarly, we use the following network with an additional parameter $\boldsymbol{W}_2 \in \mathbb{R}^{d \times m}$ to approximate $\alpha_2(\boldsymbol{z}_i)$ by

$$\alpha_2(\boldsymbol{z}_i) := \alpha_2(\boldsymbol{W}_2; \boldsymbol{e}_{1,i}, \boldsymbol{e}_{2,i}) = \log\left(\varepsilon + \frac{1}{m} \sum_{j'=1}^m \exp\left(\frac{\cos(\boldsymbol{e}_{2,i}, \boldsymbol{W}_{2,j'}) - \boldsymbol{e}_{1,i}^T \boldsymbol{e}_{2,i}}{\tau}\right)\right). \quad (11)$$

Finally, our unified objective for learning the encoders and the NPN becomes

$$\min_{\boldsymbol{w}, \tau, \boldsymbol{W}_1, \boldsymbol{W}_2} \tau \cdot \frac{1}{|\mathcal{S}|} \sum_{\boldsymbol{x}_i \in \mathcal{S}} (\exp(-\alpha_1(\boldsymbol{W}_1, \boldsymbol{e}_{1,i}, \boldsymbol{e}_{2,i})) \cdot (\varepsilon + g_1(\boldsymbol{w}, \tau; i, \mathcal{S})) + \alpha_1(\boldsymbol{W}_1, \boldsymbol{e}_{1,i}, \boldsymbol{e}_{2,i})) +$$

$$\tau \cdot \frac{1}{|\mathcal{S}|} \sum_{\boldsymbol{z}_i \in \mathcal{S}} (\exp(-\alpha_2(\boldsymbol{W}_2, \boldsymbol{e}_{1,i}, \boldsymbol{e}_{2,i})) \cdot (\varepsilon + g_2(\boldsymbol{w}, \tau; i, \mathcal{S})) + \alpha_2(\boldsymbol{W}_2, \boldsymbol{e}_{1,i}, \boldsymbol{e}_{2,i})) + 2\tau(\rho - 1).$$

$$(12)$$

### 4.3 Alternating optimization and acceleration

To solve the problem (12), a straightforward approach is to update $\boldsymbol{w}, \tau, \boldsymbol{W}_1, \boldsymbol{W}_2$ simultaneously by using stochastic gradient based methods. However, we find that this approach does not work well in practice (see Appendix B.3 for empirical results). The reasons are multi-fold: (i) the overall objective landscape of $\boldsymbol{w}, \tau, \boldsymbol{W}_1, \boldsymbol{W}_2$ is much more complicated than the original objective in terms of $\boldsymbol{w}, \tau$; (ii) the NPNs' predictions also rely on the output embeddings of the encoders, which makes the predicted normalizers from one step update of $\boldsymbol{W}_1, \boldsymbol{W}_2$ not good enough for updating the parameters $\boldsymbol{w}, \tau$. A natural idea to address this issue is to split the parameters $\boldsymbol{w}, \tau, \boldsymbol{W}_1, \boldsymbol{W}_2$ into two blocks $(\boldsymbol{w}, \tau)$ and $(\boldsymbol{W}_1, \boldsymbol{W}_2)$, and use an alternating optimization scheme to update two blocks one by one. The method is proved to enjoy a convergence guarantee (Grippof & Sciandrone, 1999, Theorem 6.3) and similar strategies have been used for other problems that exhibit two natural blocks,

---

**Algorithm 1:** The NeuCLIP Algorithm

---

**Input:** CLIP model $\boldsymbol{w}^{(0)}$, temperature $\tau^{(0)}$, NPNs $\boldsymbol{W}_1^{(0)}, \boldsymbol{W}_2^{(0)}$, dataset $\mathcal{S}$, number of iterations $T$, restart frequency $T_r$ and number of updates $T_u$ for NPNs

1 **for** $t = 0, \ldots, T - 1$ **do**
2      Randomly sample a mini-batch $\mathcal{B}^{(t)} \subset \mathcal{S}$;
3      **if** $t \mod T_r = 0$ **then**                `// Restart`
4          Reset $\boldsymbol{W}_1^{(t)}, \boldsymbol{W}_2^{(t)}$ with $\{e_{2,i}\}_{i \in \mathcal{B}^{(t)}}$ and $\{e_{1,i}\}_{i \in \mathcal{B}^{(t)}}$, respectively ;
5      Set $\boldsymbol{W}_1^{(t,0)} = \boldsymbol{W}_1^{(t)}$ and $\boldsymbol{W}_2^{(t,0)} = \boldsymbol{W}_2^{(t)}$;
6      **for** $t' = 0, \ldots, T_u - 1$ **do**             `// Multiple updates`
7          Compute mini-batch gradient of the objective (12) w.r.t. $\boldsymbol{W}_1^{(t,t')}, \boldsymbol{W}_2^{(t,t')}$;
8          Update $\boldsymbol{W}_1^{(t,t'+1)}, \boldsymbol{W}_2^{(t,t'+1)}$ with an optimizer ;
9      Set $\boldsymbol{W}_1^{(t+1)} = \boldsymbol{W}_1^{(t,T_u)}$ and $\boldsymbol{W}_2^{(t+1)} = \boldsymbol{W}_2^{(t,T_u)}$;
10      Compute $\alpha_1^{(t+1)}(\boldsymbol{x}_i)$ and $\alpha_2^{(t+1)}(\boldsymbol{z}_i)$ for $(\boldsymbol{x}_i, \boldsymbol{z}_i) \in \mathcal{B}^{(t)}$;
11      Compute mini-batch gradient of the objective (12) w.r.t $\boldsymbol{w}^{(t)}, \tau^{(t)}$;
12      Update $\boldsymbol{w}^{(t+1)}$ and $\tau^{(t+1)}$ with an optimizer;

---

e.g., non-negative matrix factorization (Lin, 2007). However, the method is not implementable as exactly solving the optimization problem over one block given another block is unrealistic. To address this, we apply only one or several stochastic gradient updates to approximately minimize the problem over each block. We present a practical method in Algorithm 1, which is referred to as NeuCLIP. For comparison, we present FastCLIP in Algorithm 2 in Appendix A.3.

**Acceleration.** We develop two techniques to accelerate training. First, we perform **multiple NPN updates** before updating the CLIP model. To avoid additional cost, we use the same batch of data from the given iteration for updating the NPNs. Since each update of the encoders changes the loss landscape of the NPNs, multiple updates enables the NPNs to maintain the same pace as the encoders and produce more accurate normalizers. In practice, we find that a small number of updates (e.g., $T_u = 10$) is sufficient. Since the NPNs are lightweight networks, the additional cost is minimal (cf. Appendix B.4 for empirical results). Second, we apply **periodic re-initialization** of the NPNs by using randomly sampled text embeddings $\{e_{2,i}\}$ to reset $\boldsymbol{W}_1$ and their corresponding image embeddings $\{e_{1,i}\}$ to reset $\boldsymbol{W}_2$. This also helps mitigate the convergence gap between the CLIP model and the NPNs. This procedure is motivated by the observation that $\boldsymbol{W}_1$ and $\boldsymbol{W}_2$ act as compact summaries of all text and image embeddings. Together, these two techniques ensure that the NPNs remain well-aligned with the evolving encoders, leading to more effective training.

**Convergence of NeuCLIP.** We analyzed the convergence property of an abstract algorithmic framework of Algorithm 1 in Algorithm 4 (cf. Appendix C for details). Let $f(\boldsymbol{w}, \tau, \boldsymbol{W}_1, \boldsymbol{W}_2)$ denote the function considered in Equation (12), in Theorem 2 we show that after $T = \mathcal{O}(\varepsilon^{-4})$ iterations, we can find an $\varepsilon$-stationary point such that $\frac{1}{T} \sum_{t=0}^{T-1} \mathbb{E}\left[\left\|\nabla_{\boldsymbol{w},\tau} f(\boldsymbol{w}^{(t)}, \tau^{(t)}, \boldsymbol{W}_1^{(t)}, \boldsymbol{W}_2^{(t)})\right\|^2\right] \leq \varepsilon^2$ under proper assumptions.

## 5 EXPERIMENTS

**Experiment Settings**. In all the experiments, we train a CLIP model on an image-text dataset with a given compute budget (i.e., number of samples to be processed) using 8 NVIDIA H100 GPUs. The text encoder of the CLIP model is a Transformer (Vaswani et al., 2017), and the image encoder is either a ViT (Dosovitskiy et al., 2021) or a ResNet (He et al., 2016). We consider five datsets, including CC3M (Changpinyo et al., 2021), CC12M (Sharma et al., 2018) and three subsets of the DFN dataset at different scales (Fang et al., 2023a), ranging from 14M to 192M and 1B. The details of the experiment settings, including batch size and training budget for each dataset, can be found in Table 1. Ablation studies are conducted on the CC3M dataset and the DFN-14M dataset.

**Evaluation Metrics**. Throughout the section, we evaluate the performance of trained models on zero-shot classification and retrieval tasks. Specifically, we leverage the Datacomp benchmark (Gadre et al., 2023) and report the average performance on its 38 tasks (denoted as Datacomp

Table 1: Details of experiment settings. "Size" denotes the number of image-text pairs we successfully downloaded. "Samples" denotes the number of samples that are processed for training, which equals the total number of iterations times global batch size. "Batch Size" denotes the global batch size.

| Dataset | Dataset Size | Samples Seen | Vision Encoder | Batch Size |
|---------|-------------|-------------|---------------|-----------|
| CC3M | 2.7M | 100M | ResNet-50 | 1,024 |
| CC12M | 9.2M | 300M | ViT-B/32 | 2,048 |
| DFN-14M | 13.7M | 320M | ViT-B/32 | 4,096 |
| DFN-192M | 192M | 1.3B | ViT-B/16 | 5,120 |
| DFN-1B | 1.0B | 3.0B | ViT-B/16 | 5,120 |

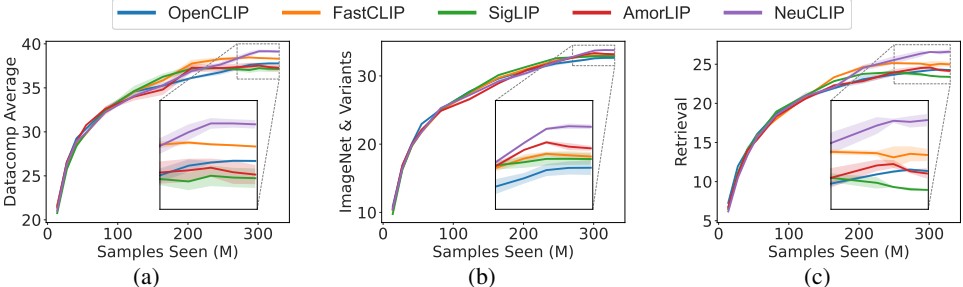

Figure 1: Performance curves of different methods on DFN-14M. (a): Datacomp Average performance. (b): ImageNet & Variants performance. (c): Retrieval performance.

Average). Moreover, we report the average performance on two subsets of the 38 tasks: (1) ImageNet & Variants, which consists of classification tasks on ImageNet-related datasets; (2) Retrieval, which consists of retrieval tasks. More information can be found in Appendix B.2.

**Hyperparameters**. For the NPNs, we set the number of hidden neurons as $m = 4,096$, restart frequency $T_r = 500$, and number of updates per iteration $T_u = 10$. We use the AdamW optimizer (Loshchilov & Hutter, 2019) to train the CLIP model and the AdaGrad optimizer (Duchi et al., 2011) to train the NPNs. We provide more details on other hyperparameters in Appendix B.1.

## 5.1 Comparison with Baselines

In this subsection, we provide comparison between our proposed NeuCLIP and several strong baselines, including OpenCLIP (Cherti et al., 2023), FastCLIP (Wei et al., 2024), SigLIP (Zhai et al., 2023) and AmorLIP (Sun et al., 2025). For OpenCLIP and SigLIP, we use the implementation from open_clip (Ilharco et al., 2021). For FastCLIP and AmorLIP, we use their released code, respectively. For experiments on CC3M, CC12M and DFN-14M, we repeat each method three times with different random seeds and report the mean. The Datacomp Average performance of different methods on the different datasets are presented in Table 2, and the full evaluation results are deferred to Appendix B.5. Additionally, we plot the performance curves during training in Figure 1.

The first observation we have is that NeuCLIP outperforms all other methods on all datasets, indicating the effectiveness of our approach. Secondly, from Figure 1 we find that NeuCLIP achieves larger improvement at the later stage of training. Note that in Algorithm 1, we optimize the NPNs $\boldsymbol{W}_1^{(t)}, \boldsymbol{W}_2^{(t)}$ given a fixed CLIP model $\boldsymbol{w}^{(t)}, \tau^{(t)}$. At later stage of training, the change in $\boldsymbol{w}^{(t)}, \tau^{(t)}$ becomes smaller, enabling the learning of NPNs to be more efficient for the updated encoders. Third, for AmorLIP, we observe differences between our results on CC3M and CC12M and those reported in Sun et al. (2025). This is because we reran the AmorLIP training on the same datasets used for the other methods, whereas our CC3M and CC12M datasets differ from those in Sun et al. (2025), as they come from different downloaded snapshots. (cf. Appendix B.5).

Table 2: Datacomp Average performance of different methods trained on different datasets.

| Method | CC3M | CC12M | DFN-14M | DFN-192M | DFN-1B |
|--------|------|-------|---------|----------|--------|
| OpenCLIP | 21.84 | 27.91 | 37.78 | 54.58 | 56.25 |
| FastCLIP | 24.74 | 31.50 | 38.45 | 54.72 | 56.68 |
| SigLIP | 22.19 | 28.60 | 37.23 | 54.26 | 56.32 |
| AmorLIP | 22.89 | 29.86 | 37.53 | 53.83 | 56.24 |
| NeuCLIP | **25.08** | **31.89** | **39.16** | **54.90** | **57.34** |

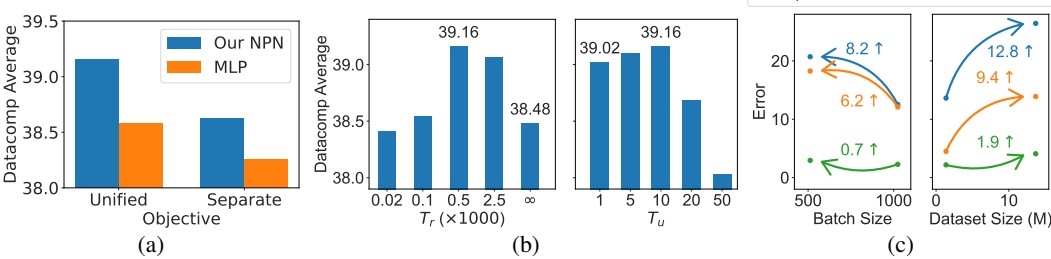

Figure 2: (a): Ablation study of training objective and architecture of NPNs. (b): Ablation study of restart frequency of NPNs (left) and number of updates (right). (c): Estimation error of NPNs.

## 5.2 ABLATION STUDY

In this subsection, we conduct ablation study of different components in NeuCLIP. We run all the experiments on DFN-14M or CC3M, where the setting is the same as in Table 1.

**Comparison with AmorLIP's Design**. The main differences between the design of AmorLIP and NeuCLIP lie in the training objective and model architecture of the NPN: (1) AmorLIP employs two separate objectives to train the CLIP model and the NPNs respectively, while we leverage a unified objective. Specifically, AmorLIP uses the following objective to train their NPNs:

$$\frac{1}{2|\mathcal{S}|} \sum\nolimits_{\boldsymbol{x}_i, \boldsymbol{z}_i \in \mathcal{S}} \left( \|\alpha_{1,i} - \log(\varepsilon + g_1(\boldsymbol{w}, \tau; i, \mathcal{S}))\|^2 + \|\alpha_{2,i} - \log(\varepsilon + g_2(\boldsymbol{w}, \tau; i, \mathcal{S}))\|^2 \right),$$

where $\alpha_{1,i}$ is the predicted log-normalizer for $\boldsymbol{x}_i$ and $\alpha_{2,i}$ is the predicted log-normalizer for $\boldsymbol{z}_i$. (2) AmorLIP chooses Multi-Layer Perceptrons (MLPs) as their NPNs, while in NeuCLIP we use single-layered NPNs that take advantage of the inductive bias in the optimal solutions of $\alpha$. In order to provide a comparison between these design choices, we conduct the following experiments: (1) In Line 7 of Algorithm 1, we compute the gradient of the NPNs using the above equation, in which case the objectives for training the CLIP model and the NPNs are not unified anymore. (2) We instantiate the NPNs with MLPs, and initialize them with random weights, which follows the practice of Sun et al. (2025). We present the Datacomp Average performance of different objectives and architectures in Figure 2a. From the results we can observe that learning with the unified objective yields better performance than learning with two separate objectives, and our inductive-biased NPN design outperforms MLPs. We also conduct the same experiments on CC3M and CC12M. The results, along with full results on DFN-14M, are presented in Tables 14 to 16, where we reach a similar conclusion.

**Restart Frequency**. To investigate the impact of the restart frequency $T_r$, we conduct experiments with different values of $T_r$, where $T_r = \infty$ means the NPNs are never re-initialized. We plot the Datacomp Average performance of different $T_r$ in the left part of Figure 2b, and present the full evaluation results in Table 17 in Appendix B.6. From the results we can observe that when $T_r > 500$, the performance decreases. Also, $T_r = 500$ gives better performance than no re-initialization (i.e., $T_r = \infty$). This is probably because the NPNs lack behind the updated encoders, making their estimations less accurate for updated encoders. Moreover, when $T_r$ is small, the NPNs are frequently set to the mini-batch features. In this case the output of the NPNs is close to the mini-batch estimators, which also leads to degraded performance.

**Multiple NPN Updates**. Another strategy to mitigate the gap of convergence speed between the NPNs and the encoders is to update the NPNs multiple ($T_u$) times before updating the encoders.

Specifically, we use the same batch of data to compute the stochastic gradient w.r.t. the NPNs' parameters across multiple updates to avoid expensive forward passes of the CLIP model. We conduct experiments with different $T_u$ and plot the results in the right part of Figure 2b. From the results we can see that as the number of updates increases, the performance first increases, and starts to decrease when $T_u > 10$. The decrease is expected since we are using the same batch of data to update the NPNs, which will overfit to the batch and provide inaccurate estimation for other samples.

**Estimation Error of Normalizers.** From Section 3 we know the estimation error of FastCLIP increases when the dataset size increases or when the batch size decreases. To demonstrate the effectiveness of our approach, we compare the estimation error of normalizers in OpenCLIP, FastCLIP, and NeuCLIP under the following two settings. Firstly, we run each method on CC3M using two batch sizes (512 and 1024). For each run, we select five checkpoints such that the corresponding checkpoints across batch size settings have seen the same number of samples. At each checkpoint, we compute the estimation error as the mean squared error between the logarithm of the predicted normalizers and the true normalizers, and then report the mean across checkpoints. More details on the computation of the estimation error are presented in Appendix B.6. As shown on the left part of Figure 2c, the error of NeuCLIP increases only marginally when the batch size decreases, whereas OpenCLIP and FastCLIP exhibit a much larger increase. Secondly, we run each method on two datasets of different sizes: a subset of DFN-14M ($n = 1.37M$) and full DFN-14M ($n = 13.7M$). On the right part of Figure 2c, we plot the average error of five checkpoints selected using the same procedure as above. The results show that NeuCLIP is only slightly affected by the increase in dataset size, in contrast to OpenCLIP and FastCLIP that suffer significant degradation.

## 6 CONCLUSION

In this paper, we have studied the problem of efficiently approximating the normalizers in the contrastive loss for training CLIP models. We proposed a novel objective that allows us to jointly learn CLIP encoders and compact networks that predict the log-normalizers of image and text data. We proposed an alternating optimization algorithm to learn the encoders and the compact networks efficiently. We conducted extensive experiments to demonstrate the effectiveness of our algorithm, and reveal insights on training of the network through ablation study.

ACKNOWLEDGMENTS

We thank anonymous reviewers for constructive comments. This work used GPU resources at TAMU ACES and NCSA Delta through allocation CIS230245 from the Advanced Cyberinfrastructure Coordination Ecosystem: Services & Support (ACCESS) program, which is supported by U.S. National Science Foundation grants #2138259, #2138286, #2138307, #2137603, and #2138296. Xiyuan Wei and Tianbao Yang were partially supported by National Science Foundation award #2306572. Tianbao Yang was partially supported by Stephen Horn'79 Engineering Excellence award at Texas A&M University. Chih-Jen Lin was supported in part by National Science and Technology Council of Taiwan grants NSTC-113-2222-E-002-005-MY3 and NSTC-114-2634-F-002-007.

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

# A TECHNICAL DETAILS

## A.1 DERIVATION OF OUR NEW OBJECTIVE AND THE OPTIMAL SOLUTION

When $f(\cdot) = -\log(\cdot)$, we have

$$f^*(y) = \max_x y \cdot x - f(x) = \max_x y \cdot x + \log(x).$$

From the first-order optimality condition we know $y + 1/x^* = 0$, which is $x^* = -1/y$. Substituting the optimal solution of $x$ into the definition of $f^*(y)$, we get $f^*(y) = -\log(-y) - 1$. Since $-\log(\cdot)$ is a convex function, we have

$$f(x) = f^{**}(x) = \max_y x \cdot y - f^*(y) = \max_y x \cdot y + \log(-y) + 1.$$

Plugging in $x = \varepsilon + g_1(\boldsymbol{w}, \tau, i, \mathcal{S})$ into the above equation, we get

$$
\begin{aligned}
F(\boldsymbol{w}, \tau; \boldsymbol{x_i}) &= \log(\varepsilon + g_1(\boldsymbol{w}, \tau; i, \mathcal{S})) \\
&= -1 \cdot (-\log(\varepsilon + g_1(\boldsymbol{w}, \tau; i, \mathcal{S}))) \\
&= -\max_y \{y \cdot (\varepsilon + g_1(\boldsymbol{w}, \tau; i, \mathcal{S})) + \log(-y) + 1\} \\
&= \min_y \{-y \cdot (\varepsilon + g_1(\boldsymbol{w}, \tau; i, \mathcal{S})) - \log(-y) - 1\} \\
&= \min_\alpha \{\exp(-\alpha) \cdot (\varepsilon + g_1(\boldsymbol{w}, \tau; i, \mathcal{S})) + \alpha - 1\},
\end{aligned}
$$

where the last equality uses a change of variable $\alpha = -\log(-y)$. This completes the derivation of the new objective. Moreover, to derive the optimal solution of $\alpha$, we can leverage the first-order optimality:

$$-\exp(-\alpha^*) \cdot (\varepsilon + g_1(\boldsymbol{w}, \tau; i, \mathcal{S})) + 1 = 0,$$

which gives $\alpha^* = \log(\varepsilon + g_1(\boldsymbol{w}, \tau; i, \mathcal{S}))$.

## A.2 INDUCING THE $u$ UPDATE OF SOGCLR FROM EQUATION (6)

Indeed, the $u$ sequence update in Equation (3) and the gradient estimator of $\boldsymbol{w}$ in Equation (4) can be derived for optimizing Equation (6). To illustrate this, we consider a stochastic block mirror descent update of $\bar{y}_{1,i} = \exp(-\alpha_{1,i})$ and $\bar{y}_{2,i} = \exp(-\alpha_{2,i})$ for $(\boldsymbol{x_i}, \boldsymbol{z_i}) \in \mathcal{B}^t$ at the $t$-th iteration:

$$\bar{y}_{1,i}^{(t+1)} = \arg\min_{y_{1,i}} y_{1,i} \cdot (\varepsilon + g_1(\boldsymbol{w}^{(t)}, \tau^{(t)}; i, \mathcal{B}^{(t)})) - \log(y_{1,i}) + \frac{1}{\eta} \cdot D(y_{1,i}, \bar{y}_{1,i}^{(t)}),$$

$$\bar{y}_{2,i}^{(t+1)} = \arg\min_{y_{2,i}} y_{2,i} \cdot (\varepsilon + g_2(\boldsymbol{w}^{(t)}, \tau^{(t)}; i, \mathcal{B}^{(t)})) - \log(y_{2,i}) + \frac{1}{\eta} \cdot D(y_{2,i}, \bar{y}_{2,i}^{(t)}),$$

where $D(u, v) = -\log(u) + \log(v) + \frac{1}{v}(u - v)$ is the Bregman divergence induced by $-\log(\cdot)$ (also known as the Itakura–Saito distance), and $\eta > 0$ is a hyperparameter. We can derive closed-form updates of $\bar{y}_{1,i}^{(t+1)}, \bar{y}_{2,i}^{(t+1)}$ as follows. By the first-order optimality condition, we have

$$\left(\varepsilon + g_1(\boldsymbol{w}^{(t)}, \tau^{(t)}; i, \mathcal{B}^{(t)}) - \frac{1}{\bar{y}_{1,i}^{(t+1)}}\right) + \frac{1}{\eta}\left(-\frac{1}{\bar{y}_{1,i}^{(t+1)}} + \frac{1}{\bar{y}_{1,i}^{(t)}}\right) = 0.$$

Rearranging the terms, we get

$$\frac{1}{\bar{y}_{1,i}^{(t+1)}} = \frac{1}{1+\eta} \cdot \frac{1}{\bar{y}_{1,i}^{(t)}} + \frac{\eta}{1+\eta} \cdot \left(\varepsilon + g_1(\boldsymbol{w}^{(t)}, \tau^{(t)}; i, \mathcal{B}^{(t)})\right),$$

which can be mapped to Equation (3) with $u_{1,i}^{(t)} = 1/\bar{y}_{1,i}^{(t)}$ and $\gamma = \eta/(1+\eta)$. With $\bar{y}_{1,i}^{(t+1)}, \bar{y}_{2,i}^{(t+1)}$, the stochastic gradient of Equation (6) w.r.t. $\boldsymbol{w}^{(t)}$ is given by

$$\tau^{(t)} \cdot \frac{1}{|\mathcal{B}^{(t)}|} \sum_{i \in \mathcal{B}^{(t)}} \bar{y}_{1,i}^{(t+1)} \cdot \nabla_{\boldsymbol{w}} g_1(\boldsymbol{w}^{(t)}, \tau^{(t)}; i, \mathcal{B}^{(t)}) + \tau^{(t)} \cdot \frac{1}{|\mathcal{B}^{(t)}|} \sum_{i \in \mathcal{B}^{(t)}} \bar{y}_{2,i}^{(t+1)} \cdot \nabla_{\boldsymbol{w}} g_2(\boldsymbol{w}^{(t)}, \tau^{(t)}; i, \mathcal{B}^{(t)}),$$

which is exactly same as in Equation (4) with $u_{1,i}^{(t)} = 1/\bar{y}_{1,i}^{(t)}$.

---

**Algorithm 2:** The FastCLIP Algorithm

---

**Input:** Model $\boldsymbol{w}^{(0)}$, temperature $\tau^{(0)}$, estimators $\boldsymbol{u}_1, \boldsymbol{u}_2$, dataset $\mathcal{S}$, number of iterations $T$

1 **for** $t = 0, \ldots, T - 1$ **do**

2      Randomly sample a mini-batch $\mathcal{B}^{(t)} \subset \mathcal{S}$;

3      Update $u_{1,i}^{(t+1)}$ and $u_{2,i}^{(t+1)}$ using Equation (3) for $i \in \mathcal{B}^{(t)}$;

4      Set $u_{1,i}^{(t+1)} = u_{1,i}^{(t)}$ and $u_{2,i}^{(t+1)} = u_{2,i}^{(t)}$ for $i \notin \mathcal{B}^{(t)}$;

5      Compute gradient estimators for $\boldsymbol{w}^{(t)}, \tau^{(t)}$ using Equations (4) and (13), respectively

$$
\frac{1}{|\mathcal{B}^{(t)}|} \sum_{i \in \mathcal{B}^{(t)}} \log(u_{1,i}^{(t+1)}) + \frac{1}{|\mathcal{B}^{(t)}|} \sum_{i \in \mathcal{B}^{(t)}} \log(u_{2,i}^{(t+1)}) + 2\rho
$$

$$
+ \tau^t \cdot \frac{1}{|\mathcal{B}^{(t)}|} \sum_{i \in \mathcal{B}^{(t)}} \frac{1}{u_{1,i}^{(t+1)}} \cdot \nabla_\tau g_1(\boldsymbol{w}^{(t)}, \tau^{(t)}; i, \mathcal{B}^{(t)}) \tag{13}
$$

$$
+ \tau^{(t)} \cdot \frac{1}{|\mathcal{B}^{(t)}|} \sum_{i \in \mathcal{B}^{(t)}} \frac{1}{u_{2,i}^{(t+1)}} \cdot \nabla_\tau g_2(\boldsymbol{w}^{(t)}, \tau^{(t)}; i, \mathcal{B}^{(t)}).
$$

6      Update $\boldsymbol{w}^{(t+1)}$ and $\tau^{(t+1)}$ using an optimizer;

---

Table 3: Hyperparameters for training the CLIP model.

| Dataset | lr | lr of $\tau$ | wd | warmup | $\rho$ |
|---------|-----|----------|-----|--------|--------|
| CC3M | 1e-3 | 1.25e-4 | 0.1 | 10,000 | 6.5 |
| CC12M | 4e-4 | 5e-5 | 0.1 | 10,000 | 8.5 |
| DFN-14M | 5e-4 | 6.25e-5 | 0.2 | 500 | 11.0 |
| DFN-192M | 3.125e-4 | 3.9e-5 | 0.2 | 500 | 11.0 |
| DFN-1B | 3.125e-4 | 3.9e-5 | 0.2 | 500 | 11.0 |

### A.3 DETAILS OF THE FASTCLIP ALGORITHM

The FastCLIP algorithm (Wei et al., 2024) is presented in Algorithm 2.

## B ADDITIONAL EXPERIMENT RESULTS

### B.1 DETAILS OF EXPERIMENT SETTINGS

**Hyperparameters**. In Table 3, we present hyperparameters for training the CLIP model. The hyperparameters are selected according to Datacomp Average performance. For each dataset, we tune the learning rate (lr) between 1e-4 and 1e-3, and tune the temperature parameter's lr between 1/8 and 1 of the model weights' lr. We set the weight decay (wd) following previous works (Gadre et al., 2023; Wei et al., 2024). We apply warmup to the lr at the beginning of training, by linearly increasing the lr from 0 to peak value across a given number of iterations (as specified by the warmup hyperparamter). For CC3M and CC12M, we set the value of $\rho$ following Wei et al. (2024). For the DFN datasets, we tune $\rho$ between 8.0 and 13.0. In Table 4, we present hyperparameters for training the NPNs. We tune the learning rate between 1e-4 and 100.0 and the weight decay between 0.0 and 1.0. We tune the number of prototypes $m$ between 1,024 and 8,192, the restart frequency $T_r$ between 0 (i.e., no restart) and 12,500, and the number of updates $T_u$ between 1 and 50.

Table 4: Hyperparameters for training the NPNs.

| Optimizer | lr | wd | $m$ | $T_r$ | $T_u$ |
|-----------|-----|-----|-------|-------|-------|
| AdaGrad | 1.0 | 0.0 | 4,096 | 500 | 10 |

---

**Algorithm 3:** Simple (Stochastic) Gradient-based Algorithm for Solving Equation (12)

---

**Input:** Model $\boldsymbol{w}^0$, Temperature $\tau^0$, Prototypes $\boldsymbol{W}_1^0, \boldsymbol{W}_2^0$, Dataset $\mathcal{S}$, Number of iterations $T$

1 **for** $t = 0, \ldots, T - 1$ **do**
2      Randomly sample a mini-batch $\mathcal{B}^{(t)} \subset \mathcal{S}$;
3      Compute mini-batch gradient of (12) w.r.t. $\boldsymbol{w}^{(t)}, \tau^{(t)}, \boldsymbol{W}_1^{(t)}, \boldsymbol{W}_2^{(t)}$;
4      Update $\boldsymbol{w}^{(t+1)}, \tau^{(t+1)}, \boldsymbol{W}_1^{(t+1)}, \boldsymbol{W}_2^{(t+1)}$ with AdamW;

---

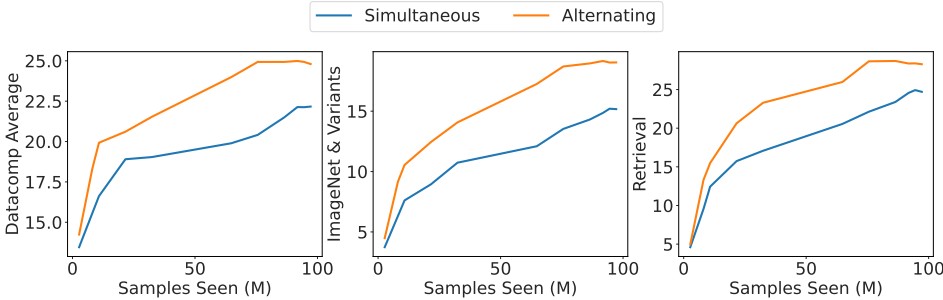

Figure 3: Comparison between simultaneous and alternating optimization.

## B.2 DETAILS OF THE DATACOMP BENCHMARK

The Datacomp Benchmark (Gadre et al., 2023) consists of 38 tasks in total, including 35 zero-shot image classification tasks and 3 zero-shot retrieval tasks. The performance metric for classification tasks is top-1 accuracy, and the performance metric for retrieval tasks is the average of image recall at 1 and text recall at 1. The "ImageNet & Variants" subset consists of ImageNet-1K (Deng et al., 2009) and 6 distribution shift datasets (Wang et al., 2019; Recht et al., 2019; Hendrycks et al., 2021a;b; Barbu et al., 2019), and the "Retrieval" subset consists of Flickr30K (Young et al., 2014) and MSCOCO (Chen et al., 2015).

## B.3 SIMULTANEOUS VS. ALTERNATING OPTIMIZATION

In this subsection, we provide comparison for two approaches solving Equation (12). The first approach is a simple gradient-based algorithm that treats all parameters as a whole and updates them simultaneously, which is presented in Algorithm 3. The second one is our NeuCLIP algorithm (Algorithm 1), which optimizes the CLIP model and the NPNs in an alternating manner. We conduct experiments on CC3M and plot the performance of the two approaches in Figure 3. From the results we can see that the joint optimization algorithm performs much worse than the alternating optimization algorithm.

## B.4 TRAINING COST OF THE NORMALIZER-PREDICTION NETWORK

In this subsection, we investigate the additional training cost incurred by the NPNs. We profile the running time of NeuCLIP with ResNet50, ViT-B/32 and ViT-B/16 as the vision encoder respectively using the PyTorch (Paszke et al., 2019) Profiler. We present the results in Table 5, from which we can observe that the additional cost of the NPNs remains low. In addition, we profile the peak memory of NeuCLIP and OpenCLIP during training. The results, reported in Table 6, suggest that memory overhead of NeuCLIP is negligible.

## B.5 COMPARISON WITH BASELINES

In this subsection, we provide more experiment results of different methods on various datasets. In Tables 7 to 11, we present the evaluation results of different methods trained on CC3M to DFN-1B.

**Variation in Performance of AmorLIP**. On CC3M and CC12M, we find that the results of our reproduction of AmorLIP is different from those reported in the AmorLIP paper. For example, their reported results of FastCLIP are lower than ours while that of AmorLIP are higher (c.f. Table 12).

Table 5: Running time of NeuCLIP under different settings. NPN Time denotes the training time of the NPNs in one iteration. Total Time denotes the running time of one iteration. Overhead denotes the portion of NPN Time in terms of Total Time.

| Vision Encoder | Embedding Dimension | NPN Time (ms) | Total Time (ms) | Overhead |
|---|---|---|---|---|
| ResNet50 | 1,024 | $49.23 \pm 1.10$ | $529.09 \pm 21.79$ | 9.30% |
| ViT-B/32 | 512 | $54.17 \pm 3.98$ | $897.79 \pm 31.89$ | 6.03% |
| ViT-B/16 | 512 | $56.43 \pm 2.83$ | $944.41 \pm 29.40$ | 5.98% |

Table 6: Memory consumption of NeuCLIP under different settings. Memory denotes the peak memory of different methods during training. Overhead denotes the memory overhead of NeuCLIP over OpenCLIP.

| Vision Encoder | Embedding Dimension | Memory (MB) | | Overhead |
|---|---|---|---|---|
| | | NeuCLIP | OpenCLIP | |
| ResNet50 | 1,024 | $10,573.2 \pm 11.10$ | $10,340.1 \pm 12.47$ | 2.25% |
| ViT-B/32 | 512 | $20,721.0 \pm 23.98$ | $20,550.0 \pm 23.47$ | 0.83% |
| ViT-B/16 | 512 | $55,171.3 \pm 33.28$ | $55,006.5 \pm 36.13$ | 0.30% |

Table 7: Evaluation results of different methods trained on CC3M. We run each method for 3 times with different random seeds, and report the average performance over 3 training runs along with standard deviation.

| Method | Datacomp Average | ImageNet & Variants | Retrieval |
|---|---|---|---|
| OpenCLIP | $21.84 \pm 0.23$ | $14.73 \pm 0.22$ | $22.25 \pm 0.38$ |
| FastCLIP | $24.74 \pm 0.35$ | $19.09 \pm 0.20$ | $29.56 \pm 0.25$ |
| SigLIP | $22.19 \pm 0.11$ | $16.08 \pm 0.16$ | $22.13 \pm 0.52$ |
| AmorLIP | $22.89 \pm 0.20$ | $17.78 \pm 0.43$ | $24.32 \pm 0.53$ |
| NeuCLIP | $\mathbf{25.08} \pm 0.39$ | $\mathbf{19.85} \pm 0.37$ | $\mathbf{30.53} \pm 0.37$ |

Table 8: Evaluation results of different methods trained on CC12M. We run each method for 3 times with different random seeds, and report the average performance over 3 training runs along with standard deviation.

| Method | Datacomp Average | ImageNet & Variants | Retrieval |
|---|---|---|---|
| OpenCLIP | $27.91 \pm 0.77$ | $21.21 \pm 0.04$ | $26.94 \pm 0.59$ |
| FastCLIP | $31.50 \pm 0.43$ | $24.61 \pm 0.05$ | $32.33 \pm 0.48$ |
| SigLIP | $28.60 \pm 0.35$ | $22.67 \pm 0.05$ | $27.99 \pm 0.25$ |
| AmorLIP | $29.86 \pm 0.83$ | $23.48 \pm 0.62$ | $28.97 \pm 0.77$ |
| NeuCLIP | $\mathbf{31.89} \pm 0.15$ | $\mathbf{25.09} \pm 0.12$ | $\mathbf{32.93} \pm 0.16$ |

Table 9: Evaluation results of different methods trained on DFN-14M. We run each method for 3 times with different random seeds, and report the average performance over 3 training runs along with standard deviation.

| Method | Datacomp Average | ImageNet & Variants | Retrieval |
|---|---|---|---|
| OpenCLIP | $37.78 \pm 0.05$ | $32.65 \pm 0.20$ | $24.33 \pm 0.06$ |
| FastCLIP | $38.45 \pm 0.06$ | $33.03 \pm 0.06$ | $25.15 \pm 0.11$ |
| SigLIP | $37.23 \pm 0.10$ | $32.89 \pm 0.17$ | $23.97 \pm 0.20$ |
| AmorLIP | $37.53 \pm 0.33$ | $33.35 \pm 0.04$ | $24.58 \pm 0.19$ |
| NeuCLIP | $\mathbf{39.16} \pm 0.20$ | $\mathbf{33.79} \pm 0.07$ | $\mathbf{26.60} \pm 0.28$ |

Table 10: Evaluation results of different methods trained on DFN-192M.

| Method | Datacomp Average | ImageNet & Variants | Retrieval |
|--------|------------------|---------------------|-----------|
| OpenCLIP | 54.58 | 55.16 | 50.42 |
| FastCLIP | 54.72 | 55.44 | 50.53 |
| SigLIP | 54.26 | 55.09 | 50.49 |
| AmorLIP | 53.83 | 55.09 | **51.43** |
| NeuCLIP | **54.90** | **55.88** | 51.39 |

Table 11: Evaluation results of different methods trained on DFN-1B.

| Method | Datacomp Average | ImageNet & Variants | Retrieval |
|--------|------------------|---------------------|-----------|
| OpenCLIP | 56.25 | 57.49 | 55.27 |
| FastCLIP | 56.68 | 57.87 | 56.05 |
| SigLIP | 56.32 | 57.20 | 55.33 |
| AmorLIP | 56.24 | 57.12 | 55.23 |
| NeuCLIP | **57.34** | **58.69** | **56.71** |

We would like to note that the training datasets used by Sun et al. (2025) and us are in fact different. This is because the two datasets are distributed with their metadata only, i.e., texts and links to their corresponding images. During the downloading process, not all images would be successfully downloaded. Specifically, Sun et al. (2025) reported a size of 2,274,566 samples for CC3M and 8,059,642 samples for CC12M, while our CC3M and CC12M datasets contain 2,723,840 and 9,187,328 samples, respectively. We tune the parameters of AmorLIP and still could not achieve the reported performance, so we believe the difference in performance is due to variation in datasets. On the other hand, if we compare their reported results with the results of NeuCLIP, we still observe an improvement of NeuCLIP over AmorLIP.

**Comparison under Same Amount of Compute**. In Table 13 we present the comparison between NeuCLIP and other baselines trained for the same amount of GPU hours. The main focus is to offset the computation overhead of NPN update (c.f. Table 5). We assume OpenCLIP, FastCLIP and SigLIP consume the same amount of compute at each iteration since they only differ in loss computation from image and text features (though Wei et al. (2024) showed that FastCLIP is slightly faster than OpenCLIP). And we reduce the number of iterations of NeuCLIP by the overhead of NPN update shown in Table 5 so that the total amount of GPU hours of NeuCLIP matches that of OpenCLIP, FastCLIP and SigLIP. From Table 13 we can observe that the performance of NeuCLIP slightly decreases compared with Table 2 but still outperforms other baselines. We did not provide results of AmorLIP, but due to the additional cost of updating the small network in AmorLIP, it is mostly likely that under the same amount of GPU hours, the performance of AmorLIP is lower than that in Table 2.

## B.6 ABLATION STUDY

In this subsection, we present detailed results of the ablation study. In Tables 14 to 16, we present evaluation results of different objectives and architectures on CC3M, CC12M and DFN-14M, respectively. In Tables 17 to 19, we present the ablation results on DFN-14M of the restart frequency $T_r$, the number of updates $T_u$ and the number of prototypes $m$, respectively. In Tables 20 and 21, we present the ablation results on CC12M of the restart frequency $T_r$ and the number of prototypes

Table 12: Datacomp Average performance of FastCLIP and AmorLIP from Sun et al. (2025) and our reproduction.

| Method | Source | CC3M | CC12M |
|--------|--------|------|-------|
| FastCLIP | Sun et al. (2025) | 23.46 | 29.00 |
| FastCLIP | Ours | 24.74 | 31.50 |
| AmorLIP | Sun et al. (2025) | 24.11 | 30.66 |
| AmorLIP | Ours | 22.89 | 29.86 |

Table 13: Comparison of Datacomp Average performance of NeuCLIP trained under the same amount of compute as baselines.

| Method | CC3M | CC12M | DFN-14M | DFN-192M | DFN-1B |
|--------|------|-------|---------|----------|--------|
| OpenCLIP | 21.84 | 27.91 | 37.78 | 54.58 | 56.25 |
| FastCLIP | 24.74 | 31.50 | 38.45 | 54.72 | 56.68 |
| SigLIP | 22.19 | 28.60 | 37.23 | 54.26 | 56.32 |
| NeuCLIP | **25.06** | **31.75** | **39.16** | **54.85** | **57.28** |

$m$, respectively. In Figure 4, we plot the estimation error of normalizers of different methods at different number of samples seen. In Figure 5, we plot the Datacomp Average performance of different restart frequency $T_r$ on CC12M.

**Computation of the Estimation Error of Normalizers.** In order to obtain the estimation error of a given model on a given dataset, we first obtain the embeddings $e_{1,i}, e_{2,i}$ for $x_i, z_i$ in the whole dataset using the model, which is done by performing forward pass on all the images and texts. Then for a given image $x_i$ and a given model, its true normalizer is computed using Equation (9). Similar procedure is applied for obtaining true normalizer for a given text. Thus the true normalizer does not incur bias or variance. To obtain the estimation error of a given model, we randomly sample 10K data points, and for each data point, we compute its true normalizer and estimators from corresponding algorithms (OpenCLIP, FastCLIP and NeuCLIP).

Table 14: Ablation of training objective and model architecture on CC3M.

| Objective | Architecture | Datacomp Average | ImageNet & Variants | Retrieval |
|-----------|--------------|------------------|---------------------|-----------|
| Unified | Our NPN | **25.08** | **19.85** | **30.53** |
| Unified | MLP | 24.84 | 19.28 | 29.50 |
| Separate | Our NPN | 24.19 | 19.20 | 29.38 |
| Separate | MLP | 24.02 | 19.09 | 29.08 |

Table 15: Ablation of training objective and model architecture on CC12M.

| Objective | Architecture | Datacomp Average | ImageNet & Variants | Retrieval |
|-----------|--------------|------------------|---------------------|-----------|
| Unified | Our NPN | **31.89** | **25.09** | **32.93** |
| Unified | MLP | 31.43 | 25.04 | 32.30 |
| Separate | Our NPN | 31.05 | 25.04 | 31.34 |
| Separate | MLP | 30.94 | 24.77 | 31.12 |

Table 16: Ablation of training objective and model architecture on DFN-14M.

| Objective | Architecture | Datacomp Average | ImageNet & Variants | Retrieval |
|-----------|--------------|------------------|---------------------|-----------|
| Unified | Our NPN | **39.16** | **33.79** | **26.60** |
| Unified | MLP | 38.58 | 33.19 | 26.35 |
| Separate | Our NPN | 38.63 | 33.70 | 25.98 |
| Separate | MLP | 38.26 | 33.12 | 25.86 |

Table 17: Ablation of restart frequency $T_r$ on DFN-14M.

| $T_r$ | Datacomp Average | ImageNet & Variants | Retrieval |
|---|---|---|---|
| 20 | 38.41 | 33.14 | 26.30 |
| 100 | 38.54 | 33.37 | 26.05 |
| 500 | **39.16** | **33.79** | **26.60** |
| 2,500 | 39.06 | 33.29 | 26.25 |
| $\infty$ | 38.48 | 33.18 | 25.79 |

Table 18: Ablation of number of updates $T_u$ on DFN-14M.

| $T_u$ | Datacomp Average | ImageNet & Variants | Retrieval |
|---|---|---|---|
| 1 | 39.02 | 33.50 | **26.65** |
| 5 | 39.10 | 33.60 | 26.48 |
| 10 | **39.16** | **33.79** | 26.60 |
| 20 | 38.68 | 33.57 | 26.05 |
| 50 | 38.03 | 33.35 | 26.46 |

Table 19: Ablation of number of prototypes $m$ on DFN-14M.

| $m$ | Datacomp Average | ImageNet & Variants | Retrieval |
|---|---|---|---|
| 1,024 | 38.57 | 33.38 | 25.92 |
| 2,048 | 38.56 | 33.73 | 26.28 |
| 4,096 | 39.16 | 33.79 | **26.60** |
| 8,192 | **39.25** | **33.90** | 26.28 |

Table 20: Ablation of restart frequency $T_r$ on CC12M.

| $T_r$ | Datacomp Average | ImageNet & Variants | Retrieval |
|---|---|---|---|
| 20 | 31.59 | 24.76 | 32.74 |
| 100 | 31.64 | 24.96 | 32.80 |
| 500 | **31.89** | **25.09** | **32.93** |
| 2500 | 31.47 | 24.88 | 32.65 |
| $\infty$ | 31.22 | 24.55 | 32.32 |

Table 21: Ablation of number of prototypes $m$ on CC12M.

| $m$ | Datacomp Average | ImageNet & Variants | Retrieval |
|---|---|---|---|
| 1024 | 31.52 | 24.73 | 32.12 |
| 2048 | 31.57 | 25.00 | 32.83 |
| 4096 | 31.89 | 25.09 | **32.93** |
| 8192 | **32.12** | **25.24** | 32.85 |

---

**Algorithm 4:** Simplified Version of the NeuCLIP Algorithm for Analysis

**Input:** Initial point $(\boldsymbol{x}^{(0)}, \boldsymbol{y}^{(0)})$, step sizes $\eta_1, \eta_2$, momentum parameter $\beta$, initial momentum $\boldsymbol{v}^{(0)}$, number of iterations $T$, number of updates $K$.

1 **for** $t = 0, \ldots, T - 1$ **do**
2    Set $\boldsymbol{y}^{(t,0)} = \boldsymbol{y}^{(t)}$;
3    **for** $k = 0, \ldots, K - 1$ **do**
4      Randomly sample $\xi^{(t,k)}$;
5      Update $\boldsymbol{y}^{(t,k+1)} = \boldsymbol{y}^{(t,k)} - \eta_2 \nabla_2 f(\boldsymbol{x}^{(t)}, \boldsymbol{y}^{(t,k)}, \xi^{(t,k)})$;
6    Set $\boldsymbol{y}^{(t+1)} = \boldsymbol{y}^{(t,K)}$;
7    Randomly sample $\xi^{(t)}$;
8    Update $\boldsymbol{v}^{(t+1)} = (1 - \beta)\boldsymbol{v}^{(t)} + \beta \nabla_1 f(\boldsymbol{x}^{(t)}, \boldsymbol{y}^{(t)}, \xi^{(t)})$;
9    Update $\boldsymbol{x}^{(t+1)} = \boldsymbol{x}^{(t)} - \eta_1 \boldsymbol{v}^{(t+1)}$;

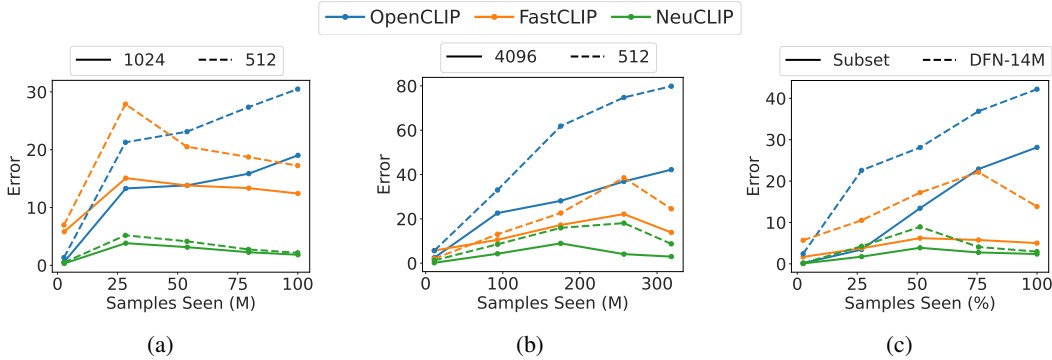

(a)          (b)          (c)

Figure 4: (a): Estimation error of normalizers of different methods with batch size 1,024 (solid lines) or 512 (dashed lines) on CC3M. (b): Estimation error of different methods with batch size 4,096 (solid lines) or 512 (dashed lines) on DFN-14M. (c): Estimation error of different methods on subset of DFN-14M (solid lines) or DFN-14M (dashed lines).

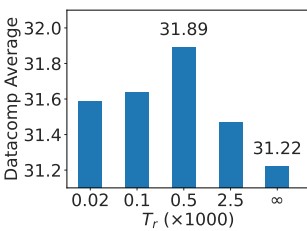

Figure 5: Ablation study of restart frequency of NPNs on CC12M.

## C   CONVERGENCE ANALYSIS OF NEUCLIP

In this section, we provide the convergence analysis of Algorithm 1. The analysis mainly follows the proof in Guo et al. (2025).

**Notations**: We let $\|\cdot\|$ denote the Euclidean norm. We use $\nabla_1$ and $\nabla_2$ to denote the gradient w.r.t. the first and second argument of a function, respectively. We use $\mathbb{E}_t[\cdot]$ to denote the expectation w.r.t. all the randomness up to iteration $t$, and we use $\mathbb{E}[\cdot]$ to denote the total expectation.

We cast the objective in Equation (12) as a function of two blocks of variables

$$f(\boldsymbol{x}, \boldsymbol{y}) := \mathbb{E}_\xi[f(\boldsymbol{x}, \boldsymbol{y}, \xi)],$$

where $\boldsymbol{x} = (\boldsymbol{w}, \tau)$, $\boldsymbol{y} = (\boldsymbol{W}_1, \boldsymbol{W}_2)$, and $\xi$ denotes a random sample. Then Equation (12) becomes $\min_{\boldsymbol{x}, \boldsymbol{y}} f(\boldsymbol{x}, \boldsymbol{y})$. We would like to show that $\boldsymbol{x}$ converges to an $\varepsilon$-stationary point of the function

$F(\boldsymbol{x}) := \min_{\boldsymbol{y}} f(\boldsymbol{x}, \boldsymbol{y})$, i.e., $\frac{1}{T} \sum_{t=0}^{T-1} \mathbb{E}[\|\nabla F(\boldsymbol{x}^{(t)})\|^2] \leq \varepsilon^2$ after $T$ iterations of optimization. Instead of directly analysing Algorithm 1, we analyze Algorithm 4 which is a slight modification of Algorithm 1 for ease of analysis:

- In Algorithm 4, we only consider using one sample to compute the stochastic gradients for both $\boldsymbol{x}$ and $\boldsymbol{y}$, instead of using mini-batches as in Algorithm 1.
- In Algorithm 4, we use Momentum SGD optimizer for $\boldsymbol{x}$.

We would like to emphasize that the analysis is not the main contribution of this work, and the analysis mainly serves as a theoretical justification of the proposed NeuCLIP method. To derive the convergence of Algorithm 4, we need the following assumptions.

**Assumption 1.** The following conditions hold:

(a) For any given $\boldsymbol{x}, \boldsymbol{y}$, let $\bar{\boldsymbol{y}}^*(\boldsymbol{x}, \boldsymbol{y}) \in \arg\min_{\boldsymbol{y}'} f(\boldsymbol{x}, \boldsymbol{y}')$ be the optimal solution closest to $\boldsymbol{y}$. It satisfies
$$\langle \boldsymbol{y} - \bar{\boldsymbol{y}}^*(\boldsymbol{x}, \boldsymbol{y}), \nabla_2 f(\boldsymbol{x}, \boldsymbol{y}) \rangle \geq \mu \|\boldsymbol{y} - \bar{\boldsymbol{y}}^*(\boldsymbol{x}, \boldsymbol{y})\|^2.$$

(b) $\nabla_1 f(\boldsymbol{x}, \boldsymbol{y})$ is $L_{11}$-Lipschitz continuous in $\boldsymbol{x}$ and $L_{12}$-Lipschitz continuous in $\boldsymbol{y}$, i.e.,
$$\|\nabla_1 f(\boldsymbol{x}, \boldsymbol{y}) - \nabla_1 f(\boldsymbol{x}', \boldsymbol{y}')\| \leq L_{11}\|\boldsymbol{x} - \boldsymbol{x}'\| + L_{12}\|\boldsymbol{y} - \boldsymbol{y}'\|, \quad \forall \boldsymbol{x}, \boldsymbol{x}', \boldsymbol{y}, \boldsymbol{y}'.$$

(c) $\nabla_2 f(\boldsymbol{x}, \boldsymbol{y})$ is $L_{21}$-Lipschitz continuous in $\boldsymbol{x}$ and $L_{22}$-Lipschitz continuous in $\boldsymbol{y}$, i.e.,
$$\|\nabla_2 f(\boldsymbol{x}, \boldsymbol{y}) - \nabla_2 f(\boldsymbol{x}', \boldsymbol{y}')\| \leq L_{21}\|\boldsymbol{x} - \boldsymbol{x}'\| + L_{22}\|\boldsymbol{y} - \boldsymbol{y}'\|, \quad \forall \boldsymbol{x}, \boldsymbol{x}', \boldsymbol{y}, \boldsymbol{y}'.$$

(d) There exist $\sigma_1, \sigma_2$ such that the stochastic gradients have bounded variance, i.e.,
$$\mathbb{E}_\xi[\|\nabla_1 f(\boldsymbol{x}, \boldsymbol{y}, \xi) - \nabla_1 f(\boldsymbol{x}, \boldsymbol{y})\|^2] \leq \sigma_1^2,$$
$$\mathbb{E}_\xi[\|\nabla_2 f(\boldsymbol{x}, \boldsymbol{y}, \xi) - \nabla_2 f(\boldsymbol{x}, \boldsymbol{y})\|^2] \leq \sigma_2^2, \quad \forall \boldsymbol{x}, \boldsymbol{y}.$$

(e) $F^* := \min_{\boldsymbol{x}} F(\boldsymbol{x}) \geq -\infty$.

**Remark 1.** Assumptions 1(b) to 1(e) are standard conditions for analyzing algorithms that optimize non-convex problems with two blocks of variables (Guo et al., 2025; Lin et al., 2025). When the models are updated in a bounded domain and have smooth activation functions, Assumptions 1(b) and 1(c) will hold. Assumption 1(a) is a mild condition that has been shown to hold for wide neural networks (Liu et al., 2023).

With the above assumptions, we get the following convergence results.

**Theorem 2.** *Under Assumption 1, let* $L_F := L_{11} + \frac{L_{12}(L_{21}+L_{22})}{\mu}$, *and setting*
$$\beta = \frac{\varepsilon^2}{5\sigma_1^2}, \quad \eta_2 = \min\left\{\frac{\mu}{L_{22}^2}, \frac{\mu\varepsilon^2}{80L_{12}^2\sigma_2^2}, \frac{1}{\mu}\right\},$$
$$\eta_1 = \min\left\{\frac{1}{2L_F}, \frac{\beta}{4L_F}, \frac{\mu^{3/2}\eta_2^{1/2}(1 - (1 - \frac{\mu\eta_2}{2})^K)^{1/2}}{8\sqrt{2}L_{12}^2(L_{12} + L_{21})}\right\},$$
$$T = \max\left\{\frac{10(F(\boldsymbol{x}^{(0)}) - F^*)}{\eta_1\varepsilon^2}, \frac{5}{\beta\varepsilon^2}\mathbb{E}\left[\left\|\boldsymbol{v}^{(0)} - \nabla F(\boldsymbol{x}^{(0)})\right\|^2\right], \frac{40L_{12}^2}{\mu\eta_2\varepsilon^2}\mathbb{E}\left[\left\|\boldsymbol{y}^{(0)} - \boldsymbol{y}^*(\boldsymbol{x}^{(0)})\right\|^2\right]\right\},$$
*then after $T$ iterations of Algorithm 4, we have*
$$\frac{1}{T}\sum_{t=0}^{T-1}\mathbb{E}\left[\left\|\nabla F(\boldsymbol{x}^{(t)})\right\|^2 + \frac{1}{4}\left\|\boldsymbol{v}^{(t)}\right\|^2\right] \leq \varepsilon^2.$$

**Corollary 1.** *Under the same assumptions as in Theorem 2, to achieve an $\varepsilon$-stationary point, i.e., $\frac{1}{T}\sum_{t=0}^{T-1}\mathbb{E}\left[\left\|\nabla F(\boldsymbol{x}^{(t)})\right\|^2\right] \leq \varepsilon^2$, the total number of stochastic gradient evaluations of $f(\boldsymbol{x}, \boldsymbol{y}, \xi)$ is $\mathcal{O}(\varepsilon^{-4})$.*

*Proof.* Ignoring the constants, we know that $T$ is in the order $\mathcal{O}(\frac{1}{\eta_1\varepsilon^2})$, $\mathcal{O}(\frac{1}{\beta\varepsilon^2})$, $\mathcal{O}(\frac{1}{\eta_2\varepsilon^2})$, whichever is the largest. The conditions of Theorem 2 imply that $\beta = \mathcal{O}(\varepsilon^2)$, $\eta_2 = \mathcal{O}(\varepsilon^2)$, and $\eta_1 = \mathcal{O}(\varepsilon^2)$. Then we know $T = \mathcal{O}(\varepsilon^{-4})$. Setting $K = \mathcal{O}(1)$, we know that the total number of stochastic gradient evaluations is $\mathcal{O}(\varepsilon^{-4})$. $\qquad\square$

**Lemma 1.** *Under Assumptions 1(a) and 1(c), the function $F(\boldsymbol{x})$ is differentiable with $\nabla F(\boldsymbol{x}) = \nabla_1 f(\boldsymbol{x}, \boldsymbol{y}^*(\boldsymbol{x}))$, where $\boldsymbol{y}^*(\boldsymbol{x})$ is any solution in $\arg\min_{\boldsymbol{y}'} f(\boldsymbol{x}, \boldsymbol{y}')$. And the gradient $\nabla F(\boldsymbol{x})$ is $L_F = L_{11} + \frac{L_{12}(L_{21}+L_{22})}{\mu}$-Lipschitz continuous.*

*Proof.* From Yuan et al. (2019, Lemma 9), we know that under Assumptions 1(a) and 1(c), the function $f(\boldsymbol{x}, \cdot)$ satisfies $\frac{\mu}{L_{22}}$-PL condition for any given $\boldsymbol{x}$. According to Nouiehed et al. (2019, Lemma A.5), we know that $F(\boldsymbol{x})$ is differentiable with $\nabla F(\boldsymbol{x}) = \nabla_1 f(\boldsymbol{x}, \boldsymbol{y}^*(\boldsymbol{x}))$, and its gradient is Lipschitz continuous with constant $L_F = L_{11} + \frac{L_{12}(L_{21}+L_{22})}{\mu}$. This completes the proof. $\quad\square$

**Lemma 2.** *Under Assumptions 1(a) to 1(d), for any given $\boldsymbol{x}, \boldsymbol{x}'$ and $\boldsymbol{y}^*(\boldsymbol{x}) \in \arg\min_{\boldsymbol{y}} f(\boldsymbol{x}, \boldsymbol{y})$, there exists $\boldsymbol{y}^*(\boldsymbol{x}') \in \arg\min_{\boldsymbol{y}} f(\boldsymbol{x}', \boldsymbol{y})$ such that*

$$\|\boldsymbol{y}^*(\boldsymbol{x}) - \boldsymbol{y}^*(\boldsymbol{x}')\| \leq \frac{L_{22}(L_{12}+L_{21})}{\mu}\|\boldsymbol{x} - \boldsymbol{x}'\|.$$

*Proof.* The proof directly follows from (Nouiehed et al., 2019, Lemma A.3) and the fact that $f(\boldsymbol{x}, \cdot)$ satisfies $\frac{\mu}{L_{22}}$-PL condition for any given $\boldsymbol{x}$ (Yuan et al., 2019, Lemma 9). $\quad\square$

**Lemma 3.** *Under Assumptions 1(b) and 1(c), for $\eta_1 \leq \frac{1}{2L_F}$, we have*

$$F(\boldsymbol{x}^{(t+1)}) \leq F(\boldsymbol{x}^{(t)}) + \frac{\eta_1}{2}\left\|\boldsymbol{v}^{(t+1)} - \nabla F(\boldsymbol{x}^{(t)})\right\|^2 - \frac{\eta_1}{2}\left\|\nabla F(\boldsymbol{x}^{(t)})\right\|^2 - \frac{\eta_1}{4}\left\|\boldsymbol{v}^{(t+1)}\right\|^2. \quad (14)$$

*Proof.* From the $L_F$-smoothness of $F$ we have

$$\begin{aligned}
F(\boldsymbol{x}^{(t+1)}) \leq& F(\boldsymbol{x}^{(t)}) + \langle \nabla F(\boldsymbol{x}^{(t)}), \boldsymbol{x}^{(t+1)} - \boldsymbol{x}^{(t)}\rangle + \frac{L_F}{2}\left\|\boldsymbol{x}^{(t+1)} - \boldsymbol{x}^{(t)}\right\|^2 \\
=& F(\boldsymbol{x}^{(t)}) - \langle \nabla F(\boldsymbol{x}^{(t)}) - \boldsymbol{v}^{(t+1)}, \boldsymbol{x}^{(t+1)} - \boldsymbol{x}^{(t)}\rangle + \langle \boldsymbol{v}^{(t+1)}, \boldsymbol{x}^{(t+1)} - \boldsymbol{x}^{(t)}\rangle \\
& + \frac{L_F}{2}\left\|\boldsymbol{x}^{(t+1)} - \boldsymbol{x}^{(t)}\right\|^2 \\
=& F(\boldsymbol{x}^{(t)}) - \eta_1 \langle \nabla F(\boldsymbol{x}^{(t)}) - \boldsymbol{v}^{(t+1)}, \boldsymbol{v}^{(t+1)}\rangle - \left(\eta_1 - \frac{L_F\eta_1^2}{2}\right)\left\|\boldsymbol{v}^{(t+1)}\right\|^2 \\
=& F(\boldsymbol{x}^{(t)}) + \eta_1\left\|\nabla F(\boldsymbol{x}^{(t)}) - \boldsymbol{v}^{(t+1)}\right\|^2 - \eta_1 \langle \nabla F(\boldsymbol{x}^{(t)}) - \boldsymbol{v}^{(t+1)}, \nabla F(\boldsymbol{x}^{(t)})\rangle \\
& - \left(\eta_1 - \frac{L_F\eta_1^2}{2}\right)\left\|\boldsymbol{v}^{(t+1)}\right\|^2.
\end{aligned}$$

Note that we have

$$\langle \nabla F(\boldsymbol{x}^{(t)}) - \boldsymbol{v}^{(t+1)}, \nabla F(\boldsymbol{x}^{(t)})\rangle = \frac{1}{2}\left(\left\|\nabla F(\boldsymbol{x}^{(t)}) - \boldsymbol{v}^{(t+1)}\right\|^2 + \left\|\nabla F(\boldsymbol{x}^{(t)})\right\|^2 - \left\|\boldsymbol{v}^{(t+1)}\right\|^2\right).$$

Thus we get

$$\begin{aligned}
F(\boldsymbol{x}^{(t+1)}) \leq& F(\boldsymbol{x}^{(t)}) + \eta_1\left\|\nabla F(\boldsymbol{x}^{(t)}) - \boldsymbol{v}^{(t+1)}\right\|^2 - \left(\eta_1 - \frac{L_F\eta_1^2}{2}\right)\left\|\boldsymbol{v}^{(t+1)}\right\|^2 \\
& - \frac{\eta_1}{2}\left(\left\|\nabla F(\boldsymbol{x}^{(t)}) - \boldsymbol{v}^{(t+1)}\right\|^2 + \left\|\nabla F(\boldsymbol{x}^{(t)})\right\|^2 - \left\|\boldsymbol{v}^{(t+1)}\right\|^2\right) \\
\leq& F(\boldsymbol{x}^{(t)}) + \frac{\eta_1}{2}\left\|\nabla F(\boldsymbol{x}^{(t)}) - \boldsymbol{v}^{(t+1)}\right\|^2 - \frac{\eta_1}{2}\left\|\nabla F(\boldsymbol{x}^{(t)})\right\|^2 - \frac{\eta_1}{4}\left\|\boldsymbol{v}^{(t+1)}\right\|^2,
\end{aligned}$$

where the last inequality is due to $\frac{L_F\eta_1^2}{2} \leq \frac{\eta_1}{4}$ by choosing $\eta_1 \leq \frac{1}{2L_F}$. This completes the proof. $\quad\square$

**Lemma 4.** *Under Assumptions 1(a) to 1(d), let* $\boldsymbol{y}^*(\boldsymbol{x}^{(t+1)})$ *be any given optimal solution in* $\arg\min_{\boldsymbol{y}'} f(\boldsymbol{x}^{(t+1)}, \boldsymbol{y}')$, *then we have*

$$
\begin{aligned}
\mathbb{E}_{t+1}\left[\left\|\boldsymbol{v}^{(t+2)} - \nabla F(\boldsymbol{x}^{(t+1)})\right\|^2\right] \leq & (1-\beta)\left\|\boldsymbol{v}^{(t+1)} - \nabla F(\boldsymbol{x}^{(t)})\right\|^2 + \frac{2L_F^2 \eta_1^2}{\beta}\left\|\boldsymbol{v}^{(t+1)}\right\|^2 \\
& + 4\beta L_{12}^2 \left\|\boldsymbol{y}^{(t+1)} - \boldsymbol{y}^*(\boldsymbol{x}^{(t+1)})\right\|^2 + \beta^2\, \sigma_1^2.
\end{aligned}
\tag{15}
$$

*Proof.* We have

$$
\begin{aligned}
& \mathbb{E}_{t+1}\left[\left\|\boldsymbol{v}^{(t+2)} - \nabla F(\boldsymbol{x}^{(t+1)})\right\|^2\right] \\
=& \mathbb{E}_{t+1}\left[\left\|(1-\beta)\boldsymbol{v}^{(t+1)} + \beta\nabla_1 f(\boldsymbol{x}^{(t+1)}, \boldsymbol{y}^{(t+1)}, \xi^{(t+1)}) - \nabla F(\boldsymbol{x}^{(t+1)})\right\|^2\right] \\
=& \mathbb{E}_{t+1}\Big[\Big\|(1-\beta)\boldsymbol{v}^{(t+1)} - \nabla F(\boldsymbol{x}^{(t+1)}) + \beta\nabla_1 f(\boldsymbol{x}^{(t+1)}, \boldsymbol{y}^{(t+1)}) \\
& \qquad + \beta(\nabla_1 f(\boldsymbol{x}^{(t+1)}, \boldsymbol{y}^{(t+1)}, \xi^{(t+1)}) - \nabla_1 f(\boldsymbol{x}^{(t+1)}, \boldsymbol{y}^{(t+1)}))\Big\|^2\Big] \\
=& \left\|(1-\beta)\boldsymbol{v}^{(t+1)} - \nabla F(\boldsymbol{x}^{(t+1)}) + \beta\nabla_1 f(\boldsymbol{x}^{(t+1)}, \boldsymbol{y}^{(t+1)})\right\|^2 \\
& + \beta^2 \mathbb{E}_{t+1}\left[\left\|\nabla_1 f(\boldsymbol{x}^{(t+1)}, \boldsymbol{y}^{(t+1)}, \xi^{(t+1)}) - \nabla_1 f(\boldsymbol{x}^{(t+1)}, \boldsymbol{y}^{(t+1)})\right\|^2\right] \\
& + 2\beta\, \mathbb{E}_{t+1}\Big[\big\langle (1-\beta)\boldsymbol{v}^{(t+1)} - \nabla F(\boldsymbol{x}^{(t+1)}) + \beta\nabla_1 f(\boldsymbol{x}^{(t+1)}, \boldsymbol{y}^{(t+1)}), \\
& \qquad\qquad \nabla_1 f(\boldsymbol{x}^{(t+1)}, \boldsymbol{y}^{(t+1)}, \xi^{(t+1)}) - \nabla_1 f(\boldsymbol{x}^{(t+1)}, \boldsymbol{y}^{(t+1)})\big\rangle\Big] \\
=& \left\|(1-\beta)\boldsymbol{v}^{(t+1)} - \nabla F(\boldsymbol{x}^{(t+1)}) + \beta\nabla_1 f(\boldsymbol{x}^{(t+1)}, \boldsymbol{y}^{(t+1)})\right\|^2 \\
& + \beta^2 \mathbb{E}_{t+1}\left[\left\|\nabla_1 f(\boldsymbol{x}^{(t+1)}, \boldsymbol{y}^{(t+1)}, \xi^{(t+1)}) - \nabla_1 f(\boldsymbol{x}^{(t+1)}, \boldsymbol{y}^{(t+1)})\right\|^2\right] \\
\leq& \left\|(1-\beta)\boldsymbol{v}^{(t+1)} - \nabla F(\boldsymbol{x}^{(t+1)}) + \beta\nabla_1 f(\boldsymbol{x}^{(t+1)}, \boldsymbol{y}^{(t+1)})\right\|^2 + \beta^2\, \sigma_1^2,
\end{aligned}
$$

where the last equality comes from the fact that

$$
\begin{aligned}
& \mathbb{E}_{t+1}\Big[\big\langle (1-\beta)\boldsymbol{v}^{(t+1)} - \nabla F(\boldsymbol{x}^{(t+1)}) + \beta\nabla_1 f(\boldsymbol{x}^{(t+1)}, \boldsymbol{y}^{(t+1)}), \\
& \qquad\qquad \nabla_1 f(\boldsymbol{x}^{(t+1)}, \boldsymbol{y}^{(t+1)}, \xi^{(t+1)}) - \nabla_1 f(\boldsymbol{x}^{(t+1)}, \boldsymbol{y}^{(t+1)})\big\rangle\Big] \\
=& \big\langle (1-\beta)\boldsymbol{v}^{(t+1)} - \nabla F(\boldsymbol{x}^{(t+1)}) + \beta\nabla_1 f(\boldsymbol{x}^{(t+1)}, \boldsymbol{y}^{(t+1)}), \\
& \quad \mathbb{E}_{t+1}[\nabla_1 f(\boldsymbol{x}^{(t+1)}, \boldsymbol{y}^{(t+1)}, \xi^{(t+1)}) - \nabla_1 f(\boldsymbol{x}^{(t+1)}, \boldsymbol{y}^{(t+1)})]\big\rangle = 0,
\end{aligned}
$$

and the last inequality is due to Assumption 1(d). Furthermore, we have

$$
\left\| (1-\beta)\boldsymbol{v}^{(t+1)} - \nabla F(\boldsymbol{x}^{(t+1)}) + \beta\nabla_1 f(\boldsymbol{x}^{(t+1)}, \boldsymbol{y}^{(t+1)}) \right\|^2
$$

$$
= \left\| (1-\beta)(\boldsymbol{v}^{(t+1)} - \nabla F(\boldsymbol{x}^{(t)})) + (1-\beta)(\nabla F(\boldsymbol{x}^{(t)}) - \nabla F(\boldsymbol{x}^{(t+1)})) \right.
$$

$$
\left. + \beta(\nabla_1 f(\boldsymbol{x}^{(t+1)}, \boldsymbol{y}^{(t+1)}) - \nabla F(\boldsymbol{x}^{(t+1)})) \right\|^2
$$

$$
\leq (1-\beta)^2(1+\beta) \left\| \boldsymbol{v}^{(t+1)} - \nabla F(\boldsymbol{x}^{(t)}) \right\|^2
$$

$$
+ \left(1 + \frac{1}{\beta}\right) \left\| (1-\beta)(\nabla F(\boldsymbol{x}^{(t)}) - \nabla F(\boldsymbol{x}^{(t+1)})) + \beta(\nabla_1 f(\boldsymbol{x}^{(t+1)}, \boldsymbol{y}^{(t+1)}) - \nabla F(\boldsymbol{x}^{(t+1)})) \right\|^2
$$

$$
\leq (1-\beta) \left\| \boldsymbol{v}^{(t+1)} - \nabla F(\boldsymbol{x}^{(t)}) \right\|^2 + \frac{2(1+\beta)(1-\beta)^2}{\beta} \left\| \nabla F(\boldsymbol{x}^{(t)}) - \nabla F(\boldsymbol{x}^{(t+1)}) \right\|^2
$$

$$
+ \frac{2(1+\beta)\beta^2}{\beta} \left\| \nabla_1 f(\boldsymbol{x}^{(t+1)}, \boldsymbol{y}^{(t+1)}) - \nabla F(\boldsymbol{x}^{(t+1)}) \right\|^2
$$

$$
\leq (1-\beta) \left\| \boldsymbol{v}^{(t+1)} - \nabla F(\boldsymbol{x}^{(t)}) \right\|^2 + \frac{2L_F^2 \eta_1^2}{\beta} \left\| \boldsymbol{v}^{(t+1)} \right\|^2 + 4\beta \left\| \nabla_1 f(\boldsymbol{x}^{(t+1)}, \boldsymbol{y}^{(t+1)}) - \nabla F(\boldsymbol{x}^{(t+1)}) \right\|^2,
$$

where the first inequality is due to the Young's inequality $\|\boldsymbol{a} + \boldsymbol{b}\|^2 \leq (1+c)\|\boldsymbol{a}\|^2 + (1+1/c)\|\boldsymbol{b}\|^2, \forall \boldsymbol{a}, \boldsymbol{b}$ and $c > 0$, and the last inequality is due to $(1-\beta)^2(1+\beta) \leq 1$ and the Lipschitz continuity of $\nabla F(\boldsymbol{x})$ (Lemma 1) and the update rule of $\boldsymbol{x}$. Thus we get

$$
\mathbb{E}_{t+1}\left[ \left\| \boldsymbol{v}^{(t+2)} - \nabla F(\boldsymbol{x}^{(t+1)}) \right\|^2 \right] \leq (1-\beta) \left\| \boldsymbol{v}^{(t+1)} - \nabla F(\boldsymbol{x}^{(t)}) \right\|^2 + \frac{2L_F^2 \eta_1^2}{\beta} \left\| \boldsymbol{v}^{(t+1)} \right\|^2
$$

$$
+ 4\beta \left\| \nabla_1 f(\boldsymbol{x}^{(t+1)}, \boldsymbol{y}^{(t+1)}) - \nabla F(\boldsymbol{x}^{(t+1)}) \right\|^2 + \beta^2 \sigma_1^2
$$

$$
\leq (1-\beta) \left\| \boldsymbol{v}^{(t+1)} - \nabla F(\boldsymbol{x}^{(t)}) \right\|^2 + \frac{2L_F^2 \eta_1^2}{\beta} \left\| \boldsymbol{v}^{(t+1)} \right\|^2
$$

$$
+ 4\beta L_{12}^2 \left\| \boldsymbol{y}^{(t+1)} - \boldsymbol{y}^*(\boldsymbol{x}^{(t+1)}) \right\|^2 + \beta^2 \sigma_1^2,
$$

where the last inequality comes from Assumption 1(b) and the fact that $\nabla F(\boldsymbol{x}^{(t+1)}) = \nabla_1 f(\boldsymbol{x}^{(t+1)}, \boldsymbol{y}^*(\boldsymbol{x}^{(t+1)}))$ for any $\boldsymbol{y}^*(\boldsymbol{x}^{(t+1)}) \in \arg\min_{\boldsymbol{y}'} f(\boldsymbol{x}^{(t+1)}, \boldsymbol{y}')$. This completes the proof. $\qquad\square$

**Lemma 5.** *Under Assumption 1, for $\eta_2 \leq \min\left\{ \frac{\mu}{L_{22}^2}, \frac{1}{\mu} \right\}$, let $\boldsymbol{y}^*(\boldsymbol{x}^{(t)})$ be any given optimal solution in $\arg\min_{\boldsymbol{y}'} f(\boldsymbol{x}^{(t)}, \boldsymbol{y}')$, then there exists $\boldsymbol{y}^*(\boldsymbol{x}^{(t+1)}) \in \arg\min_{\boldsymbol{y}'} f(\boldsymbol{x}^{(t+1)}, \boldsymbol{y}')$ such that*

$$
\mathbb{E}_t\left[ \left\| \boldsymbol{y}^{(t+1)} - \boldsymbol{y}^*(\boldsymbol{x}^{(t+1)}) \right\|^2 \right]
$$

$$
\leq \left(1 - \frac{\mu\eta_2}{2}\right)^K \left\| \boldsymbol{y}^{(t)} - \boldsymbol{y}^*(\boldsymbol{x}^{(t)}) \right\|^2 + \frac{4L_{22}^2(L_{12}+L_{21})^2\eta_1^2}{\mu^3\eta_2} \mathbb{E}_t\left[ \left\| \boldsymbol{v}^{(t+1)} \right\|^2 \right] \qquad (16)
$$

$$
+ \frac{2(1 - (1-\mu\eta_2)^K)\eta_2 \sigma_2^2}{\mu}.
$$

*Proof.* Let $\bar{\boldsymbol{y}}^*(\boldsymbol{x}, \boldsymbol{y}) \in \arg\min_{\boldsymbol{y}'} f(\boldsymbol{x}, \boldsymbol{y}')$ be the optimal solution closest to $\boldsymbol{y}$. We have

$$
\mathbb{E}_{t,k}\left[\left\|\boldsymbol{y}^{(t,k+1)} - \bar{\boldsymbol{y}}^*(\boldsymbol{x}^{(t)}, \boldsymbol{y}^{(t,k+1)})\right\|^2\right] \leq \mathbb{E}_{t,k}\left[\left\|\boldsymbol{y}^{(t,k+1)} - \bar{\boldsymbol{y}}^*(\boldsymbol{x}^{(t)}, \boldsymbol{y}^{(t,k)})\right\|^2\right]
$$

$$
=\mathbb{E}_{t,k}\left[\left\|\boldsymbol{y}^{(t,k)} - \eta_2\nabla_2 f(\boldsymbol{x}^{(t)}, \boldsymbol{y}^{(t,k)}, \xi^{(t,k)}) - \bar{\boldsymbol{y}}^*(\boldsymbol{x}^{(t)}, \boldsymbol{y}^{(t,k)})\right\|^2\right]
$$

$$
=\mathbb{E}_{t,k}\left[\left\|\boldsymbol{y}^{(t,k)} - \eta_2\nabla_2 f(\boldsymbol{x}^{(t)}, \boldsymbol{y}^{(t,k)}) - \bar{\boldsymbol{y}}^*(\boldsymbol{x}^{(t)}, \boldsymbol{y}^{(t,k)}) + \eta_2\nabla_2 f(\boldsymbol{x}^{(t)}, \boldsymbol{y}^{(t,k)})\right.\right.
$$

$$
\left.\left. -\eta_2\nabla_2 f(\boldsymbol{x}^{(t)}, \boldsymbol{y}^{(t,k)}, \xi^{(t,k)})\right\|^2\right]
$$

$$
=\left\|\boldsymbol{y}^{(t,k)} - \eta_2\nabla_2 f(\boldsymbol{x}^{(t)}, \boldsymbol{y}^{(t,k)}) - \bar{\boldsymbol{y}}^*(\boldsymbol{x}^{(t)}, \boldsymbol{y}^{(t,k)})\right\|^2
$$

$$
+ \eta_2^2\, \mathbb{E}_{t,k}\left[\left\|\nabla_2 f(\boldsymbol{x}^{(t)}, \boldsymbol{y}^{(t,k)}) - \nabla_2 f(\boldsymbol{x}^{(t)}, \boldsymbol{y}^{(t,k)}, \xi^{(t,k)})\right\|^2\right]
$$

$$
+ 2\eta_2\, \mathbb{E}_{t,k}\left[\langle \boldsymbol{y}^{(t,k)} - \eta_2\nabla_2 f(\boldsymbol{x}^{(t)}, \boldsymbol{y}^{(t,k)}) - \bar{\boldsymbol{y}}^*(\boldsymbol{x}^{(t)}, \boldsymbol{y}^{(t,k)}),\right.
$$

$$
\left. \nabla_2 f(\boldsymbol{x}^{(t)}, \boldsymbol{y}^{(t,k)}) - \nabla_2 f(\boldsymbol{x}^{(t)}, \boldsymbol{y}^{(t,k)}, \xi^{(t,k)})\rangle\right]
$$

$$
=\left\|\boldsymbol{y}^{(t,k)} - \eta_2\nabla_2 f(\boldsymbol{x}^{(t)}, \boldsymbol{y}^{(t,k)}) - \bar{\boldsymbol{y}}^*(\boldsymbol{x}^{(t)}, \boldsymbol{y}^{(t,k)})\right\|^2
$$

$$
+ \eta_2^2\, \mathbb{E}_{t,k}\left[\left\|\nabla_2 f(\boldsymbol{x}^{(t)}, \boldsymbol{y}^{(t,k)}) - \nabla_2 f(\boldsymbol{x}^{(t)}, \boldsymbol{y}^{(t,k)}, \xi^{(t,k)})\right\|^2\right]
$$

$$
\leq \left\|\boldsymbol{y}^{(t,k)} - \eta_2\nabla_2 f(\boldsymbol{x}^{(t)}, \boldsymbol{y}^{(t,k)}) - \bar{\boldsymbol{y}}^*(\boldsymbol{x}^{(t)}, \boldsymbol{y}^{(t,k)})\right\|^2 + \eta_2^2\, \sigma_2^2, \tag{17}
$$

where the first inequality holds since $\bar{\boldsymbol{y}}^*(\boldsymbol{x}^{(t)}, \boldsymbol{y}^{(t,k+1)})$ is the optimal solution closest to $\boldsymbol{y}^{(t,k+1)}$, the last equality is due to the fact that

$$
\mathbb{E}_{t,k}\left[\langle \boldsymbol{y}^{(t,k)} - \eta_2\nabla_2 f(\boldsymbol{x}^{(t)}, \boldsymbol{y}^{(t,k)}) - \bar{\boldsymbol{y}}^*(\boldsymbol{x}^{(t)}, \boldsymbol{y}^{(t,k)}),\right.
$$

$$
\left. \nabla_2 f(\boldsymbol{x}^{(t)}, \boldsymbol{y}^{(t,k)}) - \nabla_2 f(\boldsymbol{x}^{(t)}, \boldsymbol{y}^{(t,k)}, \xi^{(t,k)})\rangle\right]
$$

$$
=\langle \boldsymbol{y}^{(t,k)} - \eta_2\nabla_2 f(\boldsymbol{x}^{(t)}, \boldsymbol{y}^{(t,k)}) - \bar{\boldsymbol{y}}^*(\boldsymbol{x}^{(t)}, \boldsymbol{y}^{(t,k)}),
$$

$$
\mathbb{E}_{t,k}[\nabla_2 f(\boldsymbol{x}^{(t)}, \boldsymbol{y}^{(t,k)}) - \nabla_2 f(\boldsymbol{x}^{(t)}, \boldsymbol{y}^{(t,k)}, \xi^{(t,k)})]\rangle = 0,
$$

and the last inequality is due to Assumption 1(d). Furthermore, we have

$$
\left\|\boldsymbol{y}^{(t,k)} - \eta_2\nabla_2 f(\boldsymbol{x}^{(t)}, \boldsymbol{y}^{(t,k)}) - \bar{\boldsymbol{y}}^*(\boldsymbol{x}^{(t)}, \boldsymbol{y}^{(t,k)})\right\|^2
$$

$$
=\left\|\boldsymbol{y}^{(t,k)} - \eta_2\nabla_2 f(\boldsymbol{x}^{(t)}, \boldsymbol{y}^{(t,k)}) - \bar{\boldsymbol{y}}^*(\boldsymbol{x}^{(t)}, \boldsymbol{y}^{(t,k)}) + \eta_2\nabla_2 f(\boldsymbol{x}^{(t)}, \bar{\boldsymbol{y}}^*(\boldsymbol{x}^{(t)}, \boldsymbol{y}^{(t,k)}))\right\|^2
$$

$$
=\left\|\boldsymbol{y}^{(t,k)} - \bar{\boldsymbol{y}}^*(\boldsymbol{x}^{(t)}, \boldsymbol{y}^{(t,k)})\right\|^2 + \eta_2^2 \left\|\nabla_2 f(\boldsymbol{x}^{(t)}, \boldsymbol{y}^{(t,k)}) - \nabla_2 f(\boldsymbol{x}^{(t)}, \bar{\boldsymbol{y}}^*(\boldsymbol{x}^{(t)}, \boldsymbol{y}^{(t,k)}))\right\|^2
$$

$$
- 2\eta_2 \langle \boldsymbol{y}^{(t,k)} - \bar{\boldsymbol{y}}^*(\boldsymbol{x}^{(t)}, \boldsymbol{y}^{(t,k)}), \nabla_2 f(\boldsymbol{x}^{(t)}, \boldsymbol{y}^{(t,k)})\rangle
$$

$$
\leq \left\|\boldsymbol{y}^{(t,k)} - \bar{\boldsymbol{y}}^*(\boldsymbol{x}^{(t)}, \boldsymbol{y}^{(t,k)})\right\|^2 + \eta_2^2 \left\|\nabla_2 f(\boldsymbol{x}^{(t)}, \boldsymbol{y}^{(t,k)}) - \nabla_2 f(\boldsymbol{x}^{(t)}, \bar{\boldsymbol{y}}^*(\boldsymbol{x}^{(t)}, \boldsymbol{y}^{(t,k)}))\right\|^2
$$

$$
- 2\mu\eta_2 \left\|\boldsymbol{y}^{(t,k)} - \bar{\boldsymbol{y}}^*(\boldsymbol{x}^{(t)}, \boldsymbol{y}^{(t,k)})\right\|^2
$$

$$
\leq (1 + L_{22}^2\eta_2^2 - 2\mu\eta_2) \left\|\boldsymbol{y}^{(t,k)} - \bar{\boldsymbol{y}}^*(\boldsymbol{x}^{(t)}, \boldsymbol{y}^{(t,k)})\right\|^2 \leq (1 - \mu\eta_2) \left\|\boldsymbol{y}^{(t,k)} - \bar{\boldsymbol{y}}^*(\boldsymbol{x}^{(t)}, \boldsymbol{y}^{(t,k)})\right\|^2, \tag{18}
$$

where the first equality is due to the fact that $\nabla_2 f(\boldsymbol{x}^{(t)}, \bar{\boldsymbol{y}}^*(\boldsymbol{x}^{(t)}, \boldsymbol{y}^{(t,k)})) = \boldsymbol{0}$, the first inequality is due to Assumption 1(a), the second inequality is due to the $L_{22}$-Lipschitz continuity of $\nabla_2 f(\boldsymbol{x}, \cdot)$, and the last inequality comes from the choices of $\eta_2 \leq \frac{\mu}{L_{22}^2}$ and $\eta_2 \leq \frac{1}{\mu}$. Combining Equation (17) and Equation (18), we get

$$
\mathbb{E}_t\left[\left\|\boldsymbol{y}^{(t,k+1)} - \bar{\boldsymbol{y}}^*(\boldsymbol{x}^{(t)}, \boldsymbol{y}^{(t,k+1)})\right\|^2\right] \leq (1 - \mu\eta_2)\mathbb{E}_t\left[\left\|\boldsymbol{y}^{(t,k)} - \bar{\boldsymbol{y}}^*(\boldsymbol{x}^{(t)}, \boldsymbol{y}^{(t,k)})\right\|^2\right] + \eta_2^2\, \sigma_2^2.
$$

Telescoping over $k = 0, \ldots, K-1$, and noting that $\boldsymbol{y}^{(t)} = \boldsymbol{y}^{(t,0)}, \boldsymbol{y}^{(t+1)} = \boldsymbol{y}^{(t,K)}$, we have

$$\mathbb{E}_t \left[ \left\| \boldsymbol{y}^{(t+1)} - \bar{\boldsymbol{y}}^*(\boldsymbol{x}^{(t)}, \boldsymbol{y}^{(t+1)}) \right\|^2 \right] \leq (1-\mu\eta_2)^K \left\| \boldsymbol{y}^{(t)} - \bar{\boldsymbol{y}}^*(\boldsymbol{x}^{(t)}, \boldsymbol{y}^{(t)}) \right\|^2 + \frac{(1-(1-\mu\eta_2)^K)\eta_2\,\sigma_2^2}{\mu}. \tag{19}$$

From Lemma 2 we know there exists $\boldsymbol{y}^*(\boldsymbol{x}^{(t+1)}) \in \arg\min_{\boldsymbol{y}'} f(\boldsymbol{x}^{(t+1)}, \boldsymbol{y}')$ such that

$$\|\boldsymbol{y}^*(\boldsymbol{x}^{(t)}) - \boldsymbol{y}^*(\boldsymbol{x}^{(t+1)})\| \leq \frac{L_{22}(L_{12}+L_{21})}{\mu}\|\boldsymbol{x}^{(t)} - \boldsymbol{x}^{(t+1)}\|.$$

By considering this $\boldsymbol{y}^*(\boldsymbol{x}^{(t+1)})$, we have

$$\mathbb{E}_t \left[ \left\| \boldsymbol{y}^{(t+1)} - \boldsymbol{y}^*(\boldsymbol{x}^{(t+1)}) \right\|^2 \right]$$

$$= \mathbb{E}_t \left[ \left\| \boldsymbol{y}^{(t+1)} - \bar{\boldsymbol{y}}^*(\boldsymbol{x}^{(t)}, \boldsymbol{y}^{(t+1)}) + \bar{\boldsymbol{y}}^*(\boldsymbol{x}^{(t)}, \boldsymbol{y}^{(t+1)}) - \boldsymbol{y}^*(\boldsymbol{x}^{(t+1)}) \right\|^2 \right]$$

$$\leq \left(1 + \frac{\mu\eta_2}{2}\right) \mathbb{E}_t \left[ \left\| \boldsymbol{y}^{(t+1)} - \bar{\boldsymbol{y}}^*(\boldsymbol{x}^{(t)}, \boldsymbol{y}^{(t+1)}) \right\|^2 \right]$$

$$\quad + \left(1 + \frac{2}{\mu\eta_2}\right) \mathbb{E}_t \left[ \left\| \bar{\boldsymbol{y}}^*(\boldsymbol{x}^{(t)}, \boldsymbol{y}^{(t+1)}) - \boldsymbol{y}^*(\boldsymbol{x}^{(t+1)}) \right\|^2 \right]$$

$$\leq \left(1 + \frac{\mu\eta_2}{2}\right) \left( (1-\mu\eta_2)^K \left\| \boldsymbol{y}^{(t)} - \bar{\boldsymbol{y}}^*(\boldsymbol{x}^{(t)}, \boldsymbol{y}^{(t)}) \right\|^2 + \frac{(1-(1-\mu\eta_2)^K)\eta_2\,\sigma_2^2}{\mu} \right)$$

$$\quad + \left(1 + \frac{2}{\mu\eta_2}\right) \frac{L_{22}^2(L_{12}+L_{21})^2}{\mu^2} \mathbb{E}_t \left[ \left\| \boldsymbol{x}^{(t)} - \boldsymbol{x}^{(t+1)} \right\|^2 \right]$$

$$\leq \left(1 - \frac{\mu\eta_2}{2}\right)^K \left\| \boldsymbol{y}^{(t)} - \boldsymbol{y}^*(\boldsymbol{x}^{(t)}) \right\|^2 + \frac{4L_{22}^2(L_{12}+L_{21})^2\eta_1^2}{\mu^3\eta_2} \mathbb{E}_t \left[ \left\| \boldsymbol{v}^{(t+1)} \right\|^2 \right]$$

$$\quad + \frac{2(1-(1-\mu\eta_2)^K)\eta_2\,\sigma_2^2}{\mu},$$

where the first inequality is due to Young's inequality $\|\boldsymbol{a}+\boldsymbol{b}\|^2 \leq (1+c)\|\boldsymbol{a}\|^2 + (1+1/c)\|\boldsymbol{b}\|^2, \forall\,\boldsymbol{a},\boldsymbol{b}$ and $c > 0$, the second inequality comes from Equation (19) and Lemma 2, and the last inequality is due to the fact that $\bar{\boldsymbol{y}}^*(\boldsymbol{x}^{(t)}, \boldsymbol{y}^{(t)})$ is cloeset to $\boldsymbol{y}^{(t)}$, the update rule of $\boldsymbol{x}$, the choice that $\eta_2 \leq \frac{1}{\mu}$ and the fact that

$$\left(1 + \frac{\mu\eta_2}{2}\right)(1-\mu\eta_2)^K = \left(1 + \frac{\mu\eta_2}{2}\right)(1-\mu\eta_2)(1-\mu\eta_2)^{K-1}$$

$$\leq \left(1 - \frac{\mu\eta_2}{2}\right)(1-\mu\eta_2)^{K-1} \leq \left(1 - \frac{\mu\eta_2}{2}\right)^K.$$

This completes the proof. $\qquad\square$

Next we present the proof of Theorem 2.

*Proof of Theorem 2.* Multiplying Equation (15) by $a := \frac{\eta_1}{2\beta}$, Equation (16) by $b := \frac{2\eta_1 L_{12}^2}{1-(1-\frac{\mu\eta_2}{2})^K}$, and adding them to Equation (14), we have

$$F(\boldsymbol{x}^{(t+1)}) + a\,\mathbb{E}_{t+1} \left[ \left\| \boldsymbol{v}^{(t+2)} - \nabla F(\boldsymbol{x}^{(t+1)}) \right\|^2 \right] + b\,\mathbb{E}_t \left[ \left\| \boldsymbol{y}^{(t+1)} - \boldsymbol{y}^*(\boldsymbol{x}^{(t+1)}) \right\|^2 \right]$$

$$\leq F(\boldsymbol{x}^{(t)}) + \frac{\eta_1}{2} \left\| \boldsymbol{v}^{(t+1)} - \nabla F(\boldsymbol{x}^{(t)}) \right\|^2 - \frac{\eta_1}{2} \left\| \nabla F(\boldsymbol{x}^{(t)}) \right\|^2 - \frac{\eta_1}{4} \left\| \boldsymbol{v}^{(t+1)} \right\|^2$$

$$\quad + a(1-\beta) \left\| \boldsymbol{v}^{(t+1)} - \nabla F(\boldsymbol{x}^{(t)}) \right\|^2 + a\,\frac{2L_F^2\eta_1^2}{\beta} \left\| \boldsymbol{v}^{(t)} \right\|^2 + 4a\beta L_{12}^2 \left\| \boldsymbol{y}^{(t+1)} - \boldsymbol{y}^*(\boldsymbol{x}^{(t+1)}) \right\|^2$$

$$\quad + b \left(1 - \frac{\mu\eta_2}{2}\right)^K \left\| \boldsymbol{y}^{(t)} - \boldsymbol{y}^*(\boldsymbol{x}^{(t)}) \right\|^2 + b\frac{4L_{22}^2(L_{12}+L_{21})^2\eta_1^2}{\mu^3\eta_2} \left\| \boldsymbol{v}^{(t)} \right\|^2$$

$$\quad + a\,\beta^2\,\sigma_1^2 + b\,\frac{2(1-(1-\mu\eta_2)^K)\eta_2\,\sigma_2^2}{\mu}.$$

Taking total expectation on both sides and rearranging the terms, we have

$$\mathbb{E}[F(\boldsymbol{x}^{(t+1)})] + a\,\mathbb{E}\left[\left\|\boldsymbol{v}^{(t+2)} - \nabla F(\boldsymbol{x}^{(t+1)})\right\|^2\right] + (b - 4a\beta L_{12}^2)\,\mathbb{E}\left[\left\|\boldsymbol{y}^{(t+1)} - \boldsymbol{y}^*(\boldsymbol{x}^{(t+1)})\right\|^2\right]$$

$$\leq \mathbb{E}[F(\boldsymbol{x}^{(t)})] + \left(a(1-\beta) + \frac{\eta_1}{2}\right)\mathbb{E}\left[\left\|\boldsymbol{v}^{(t+1)} - \nabla F(\boldsymbol{x}^{(t)})\right\|^2\right]$$

$$+ b\left(1 - \frac{\mu\eta_2}{2}\right)^K \mathbb{E}\left[\left\|\boldsymbol{y}^{(t)} - \boldsymbol{y}^*(\boldsymbol{x}^{(t)})\right\|^2\right]$$

$$- \frac{\eta_1}{2}\,\mathbb{E}\left[\left\|\nabla F(\boldsymbol{x}^{(t)})\right\|^2\right] + \left(a\frac{2L_F^2\eta_1^2}{\beta} - \frac{\eta_1}{4} + b\frac{4L_{22}^2(L_{12}+L_{21})^2\eta_1^2}{\mu^3\eta_2}\right)\mathbb{E}\left[\left\|\boldsymbol{v}^{(t)}\right\|^2\right]$$

$$+ a\,\beta^2\,\sigma_1^2 + b\,\frac{2(1 - (1-\mu\eta_2)^K)\eta_2\,\sigma_2^2}{\mu}.$$

With the choices of $a, b$, we have

$$a(1-\beta) + \frac{\eta_1}{2} = \left(\frac{1}{2\beta} - \frac{1}{2} + \frac{1}{2}\right)\eta_1 = \frac{1}{2\beta}\,\eta_1 = a,$$

$$b - 4a\beta L_{12}^2 = b - 2L_{12}^2\eta_1 = b\left(1 - (1 - (1-\frac{\mu\eta_2}{2})^K)\right) = b\left(1 - \frac{\mu\eta_2}{2}\right)^K.$$

Setting

$$\Upsilon^{(t)} := \mathbb{E}[F(\boldsymbol{x}^{(t)})] + a\,\mathbb{E}\left[\left\|\boldsymbol{v}^{(t+1)} - \nabla F(\boldsymbol{x}^{(t)})\right\|^2\right] + (b - 4a\beta L_{12}^2)\,\mathbb{E}\left[\left\|\boldsymbol{y}^{(t)} - \boldsymbol{y}^*(\boldsymbol{x}^{(t)})\right\|^2\right],$$

we have

$$\Upsilon^{(t+1)} \leq \Upsilon^{(t)} - \frac{\eta_1}{2}\,\mathbb{E}\left[\left\|\nabla F(\boldsymbol{x}^{(t)})\right\|^2\right] + \left(\frac{L_F^2\eta_1^3}{\beta^2} - \frac{\eta_1}{4} + \frac{8L_{12}^4(L_{12}+L_{21})^2\eta_1^3}{\mu^3\eta_2(1-(1-\frac{\mu\eta_2}{2})^K)}\right)\mathbb{E}\left[\left\|\boldsymbol{v}^{(t)}\right\|^2\right]$$

$$+ \frac{\eta_1\beta\,\sigma_1^2}{2} + \frac{2\eta_1 L_{12}^2}{(1-(1-\frac{\mu\eta_2}{2})^K)}\cdot\frac{2(1-(1-\mu\eta_2)^K)\eta_2\,\sigma_2^2}{\mu}.$$

$$\tag{20}$$

If

$$\eta_1 \leq \frac{\beta}{4L_F}, \quad \eta_1 \leq \frac{\mu^{3/2}\eta_2^{1/2}(1-(1-\frac{\mu\eta_2}{2})^K)^{1/2}}{8\sqrt{2}L_{12}^2(L_{12}+L_{21})},$$

we have

$$\frac{L_F^2\eta_1^2}{\beta^2}\cdot\eta_1 - \frac{\eta_1}{4} + \frac{8L_{12}^4(L_{12}+L_{21})^2\eta_1^3}{\mu^3\eta_2(1-(1-\frac{\mu\eta_2}{2})^K)} \leq \frac{\eta_1}{16} - \frac{\eta_1}{4} + \frac{\eta_1}{16} = -\frac{\eta_1}{8}.$$

Plugging the above inequality into Equation (20), and rearranging the terms, we have

$$\frac{\eta_1}{2}\,\mathbb{E}\left[\left\|\nabla F(\boldsymbol{x}^{(t)})\right\|^2\right] + \frac{\eta_1}{8}\,\mathbb{E}\left[\left\|\boldsymbol{v}^{(t)}\right\|^2\right] \leq \Upsilon^{(t)} - \Upsilon^{(t+1)} + \frac{\eta_1\beta\,\sigma_1^2}{2} + \frac{4\eta_1 L_{12}^2(1-(1-\mu\eta_2)^K)\eta_2\,\sigma_2^2}{\mu(1-(1-\frac{\mu\eta_2}{2})^K)}.$$

Telescoping over $t = 0, \ldots, T-1$, and dividing both sides by $\eta_1$, we have

$$\frac{1}{T}\sum_{t=0}^{T-1}\mathbb{E}\left[\frac{1}{2}\left\|\nabla F(\boldsymbol{x}^{(t)})\right\|^2 + \frac{1}{8}\left\|\boldsymbol{v}^{(t)}\right\|^2\right] \leq \frac{\Upsilon^{(0)} - \Upsilon^{(T)}}{T\eta_1} + \frac{\beta\,\sigma_1^2}{2} + \frac{4L_{12}^2(1-(1-\mu\eta_2)^K)\eta_2\,\sigma_2^2}{\mu(1-(1-\frac{\mu\eta_2}{2})^K)}.$$

Note that with our choices of $a, b$, we have $a \geq 0, b - 4a\beta L_{12}^2 \geq 0$, which implies $\Upsilon^{(T)} \geq F^*$. Then we get

$$\frac{1}{T}\sum_{t=0}^{T-1}\mathbb{E}\left[\left\|\nabla F(\boldsymbol{x}^{(t)})\right\|^2 + \frac{1}{4}\left\|\boldsymbol{v}^{(t)}\right\|^2\right]$$

$$\leq \frac{2(\Upsilon^{(0)} - F^*)}{T\eta_1} + \beta\,\sigma_1^2 + \frac{8L_{12}^2(1-(1-\mu\eta_2)^K)\eta_2\,\sigma_2^2}{\mu(1-(1-\frac{\mu\eta_2}{2})^K)}$$

$$= \frac{2(F(\boldsymbol{x}^{(0)}) - F^*)}{T\eta_1} + \beta\,\sigma_1^2 + \frac{8L_{12}^2(1-(1-\mu\eta_2)^K)\eta_2\,\sigma_2^2}{\mu(1-(1-\frac{\mu\eta_2}{2})^K)}$$

$$+ \frac{1}{T}\cdot\frac{1}{\beta}\,\mathbb{E}\left[\left\|\boldsymbol{v}^{(0)} - \nabla F(\boldsymbol{x}^{(0)})\right\|^2\right] + \frac{1}{T}\cdot\frac{(1-\frac{\mu\eta_2}{2})^K\cdot 4L_{12}^2}{(1-(1-\frac{\mu\eta_2}{2})^K)}\,\mathbb{E}\left[\left\|\boldsymbol{y}^{(0)} - \boldsymbol{y}^*(\boldsymbol{x}^{(0)})\right\|^2\right].$$

Combining the conditions in Lemmas 3 and 5, for

$$\beta = \frac{\varepsilon^2}{5\sigma_1^2}, \quad \eta_2 = \min\left\{\frac{\mu}{L_{22}^2}, \frac{\mu\varepsilon^2}{80L_{12}^2\sigma_2^2}, \frac{1}{\mu}\right\},$$

$$\eta_1 = \min\left\{\frac{1}{2L_F}, \frac{\beta}{4L_F}, \frac{\mu^{3/2}\eta_2^{1/2}(1-(1-\frac{\mu\eta_2}{2})^K)^{1/2}}{8\sqrt{2}L_{12}^2(L_{12}+L_{21})}\right\},$$

$$T = \max\left\{\frac{10(F(\boldsymbol{x}^{(0)})-F^*)}{\eta_1\varepsilon^2}, \frac{5}{\beta\varepsilon^2}\mathbb{E}\left[\left\|\boldsymbol{v}^{(0)}-\nabla F(\boldsymbol{x}^{(0)})\right\|^2\right], \frac{40L_{12}^2}{\mu\eta_2\varepsilon^2}\mathbb{E}\left[\left\|\boldsymbol{y}^{(0)}-\boldsymbol{y}^*(\boldsymbol{x}^{(0)})\right\|^2\right]\right\},$$

we have

$$\frac{2(F(\boldsymbol{x}^{(0)})-F^*)}{T\eta_1} \le \frac{\varepsilon^2}{5}, \quad \beta\sigma_1^2 \le \frac{\varepsilon^2}{5}, \quad \frac{1}{T}\cdot\frac{1}{\beta}\mathbb{E}\left[\left\|\boldsymbol{v}^{(0)}-\nabla F(\boldsymbol{x}^{(0)})\right\|^2\right] \le \frac{\varepsilon^2}{5},$$

$$\frac{1}{T}\cdot\frac{(1-\frac{\mu\eta_2}{2})^K\cdot 4L_{12}^2}{(1-(1-\frac{\mu\eta_2}{2})^K)}\mathbb{E}\left[\left\|\boldsymbol{y}^{(0)}-\boldsymbol{y}^*(\boldsymbol{x}^{(0)})\right\|^2\right]$$

$$\le\frac{1}{T}\cdot\frac{4L_{12}^2}{(1-(1-\frac{\mu\eta_2}{2}))}\mathbb{E}\left[\left\|\boldsymbol{y}^{(0)}-\boldsymbol{y}^*(\boldsymbol{x}^{(0)})\right\|^2\right] \le \frac{\varepsilon^2}{5}.$$

Additionally, we can show that

$$\frac{1-(1-\mu\eta_2)^K}{1-(1-\frac{\mu\eta_2}{2})^K} \le 2, \forall K \ge 1, \tag{21}$$

and thus we have

$$\frac{8L_{12}^2(1-(1-\mu\eta_2)^K)\eta_2\sigma_2^2}{\mu(1-(1-\frac{\mu\eta_2}{2})^K)} \le \frac{\varepsilon^2}{5}.$$

To show Equation (21), note that the equality holds for $K=1$. Assume the inequality holds for some $K$, then for $K+1$ we have

$$\frac{1-(1-\mu\eta_2)^{K+1}}{1-(1-\frac{\mu\eta_2}{2})^{K+1}} = \frac{(1-(1-\mu\eta_2))\cdot\sum_{k=0}^K(1-\mu\eta_2)^k}{(1-(1-\frac{\mu\eta_2}{2}))\cdot\sum_{k=0}^K(1-\frac{\mu\eta_2}{2})^k}$$

$$= 2\cdot\frac{\sum_{k=0}^K(1-\mu\eta_2)^k}{\sum_{k=0}^K(1-\frac{\mu\eta_2}{2})^k} \le 2,$$

where the inequality comes from the fact that $1-\frac{\mu\eta_2}{2} \ge 0$ and $1-\mu\eta_2 \le 1-\frac{\mu\eta_2}{2}$. Combining the above inequalities, we get

$$\frac{1}{T}\sum_{t=0}^{T-1}\mathbb{E}\left[\left\|\nabla F(\boldsymbol{x}^{(t)})\right\|^2 + \frac{1}{4}\left\|\boldsymbol{v}^{(t)}\right\|^2\right] \le \varepsilon^2.$$

This completes the proof. $\qquad\square$

## D  THE USE OF LARGE LANGUAGE MODELS (LLMS)

We use LLMs to help find recent applications of CLIP models, and to help search for works that leverage auxiliary networks in other fields than CLIP training, such as Computer Vision and Natural Language Processing. We also use LLMs to help polish up writing.

