# OpenReview forum: "NeuCLIP: Efficient Large-Scale CLIP Training with Neural Normalizer Optimization"
_ICLR.cc/2026/Conference — ICLR 2026 Poster_

### Official Review · Reviewer_TYsU · 2025-10-22

**Soundness:** 3
**Presentation:** 3
**Contribution:** 3
**Rating:** 6
**Confidence:** 3

**Summary:**

This paper looks into a fundamental issue in CLIP model training and its dependence on large batch size for good performance. It looks into existing techniques that are less dependent on large batch size but their normalization approximation for log-partition estimation has room for improvement. The authors proposes a normalizer prediction network (NPN). The training objective jointly learns the encoder and a small network that predicts the log-normalizer for image and text

**Strengths:**

1. The paper has novel contribution and addresses an important issue in CLIP training. This is helpful for the research community specially when they are looking to train on a smaller budget
2. The method beats others in Datacomp. Although in some cases (table 9, 10) the improvement in marginal

**Weaknesses:**

1. Probably needs some more in related work. For example https://arxiv.org/abs/2409.13079 and https://openreview.net/pdf?id=3i13Gev2hV and how this work is tackling a different aspect of image-text pretraining
2. The paper has novel contribution but it is not clear how those improved results number actually help in retrieval. Probably some quantitative examples will be helpful
3. Minor: typo in line 58, 60. SigCLIP should be SigLIP
4. Having insight on the limitation of the proposed approach and how it can be improved further will be helpful

**Questions:**

See weakness

---

> ### Author Response · Authors · 2025-11-21
>
> We thank the reviewer for their constructive and positive evaluation of our work. We have fixed the typos the reviewer pointed out in our revised version.
>
> > **Q1**: More related works need to be added. How this work is tackling a different aspect of image-text pretraining compared with them?
>
> **A**: We thank the reviewer for pointing out additional relevant works, and we have added them in the related works. Both works focus on improving the geometry of the CLIP embedding space, such as using Euclidean (EuCLIP) or Hyperbolic (HyCoCLIP) spaces. Our work is orthogonal to this direction. We keep the conventional CLIP embedding geometry and instead address the accurate and efficient estimation of the normalization term for global contrastive loss. In principle, our NeuCLIP framework could be applied to improve the performance of EuCLIP or HyCoCLIP.
>
> > **Q2**: The paper has novel contribution but it is not clear how those improved results number actually help in retrieval. Probably some quantitative examples will be helpful
>
> **A**: We evaluated all methods on zero-shot retrieval tasks on MSCOCO and Flickr, which is part of the Datacomp benchmark. The quantitative results (average of image recall@1 and text recall@1) were reported in Tables 7 to 11 in Appendix B.5, and we also present them in the following for your convenience. From the results we can observe that NeuCLIP outperforms other baselines.
>
> | Method | CC3M | CC12M | DFN-14M | DFN-192M | DFN-1B |
> | -- | -- | -- | -- | -- | -- |
> | OpenCLIP | 22.25 | 26.94 | 24.33 | 50.42 | 55.27 |
> | FastCLIP | 29.56 | 32.33 | 25.15 | 50.53 | 56.05 |
> | SigLIP | 22.13 | 27.99 | 23.97 | 50.49 | 55.33 |
> | AmorLIP | 24.32 | 28.97 | 24.58 | **51.43** | 55.23 |
> | NeuCLIP | **30.53** | **32.93** | **26.60** | 51.39 | **56.71** |
>
> > **Q3**: Having insight on the limitation of the proposed approach and how it can be improved further will be helpful
>
> **A**: Thanks for the question. One area for improvement lies in the optimizer used for the NPN. Because the NPN is built on top of the backbone encoder, updates to the backbone make it difficult for the NPN to keep pace during training. In our paper, we used periodic re-initialization and multiple update steps to mitigate this issue. However, developing more effective optimization strategies remains an interesting open problem and could generalize to other training setups as well.

---

> > ### Comment · Reviewer_TYsU · 2025-11-23
> >
> > Thank you to the authors for their reply. I will keep the score as is.

---

### Official Review · Reviewer_fUVb · 2025-10-25

**Soundness:** 2
**Presentation:** 2
**Contribution:** 2
**Rating:** 4
**Confidence:** 4

**Summary:**

The paper proposes NeuCLIP, a training framework for CLIP that replaces per-sample moving-average normalizer estimates with a neural normalizer–prediction network (NPN) trained jointly with the encoders. The authors (1) use convex conjugate analysis to rewrite each per-sample contrastive loss as a minimization over an auxiliary log-normalizer variable, and (2) use a variational argument to replace the set of per-sample variables by a function class, then parameterize that function with compact networks operating on encoder embeddings. Training alternates between several fast NPN updates (with periodic proto-reinitialization) and encoder/temperature updates. Experiments on CC3M, CC12M, and DFN at 14M/192M/1B scales report higher Datacomp averages than OpenCLIP, FastCLIP, SigLIP, and AmorLIP under the authors’ setups; ablations examine unified vs. separate objectives, NPN architecture, restart frequency, and NPN update count, and profiling suggests single-digit percent overhead for the NPN step.

**Strengths:**

1. The loss is reformulated via a Fenchel conjugate into a per-sample minimization whose optimizer equals the log-normalizer; a variational theorem is then invoked to justify searching over functions and learning an NPN. The paper provides the derivations and the induced FastCLIP update as a special case.


2. Results span multiple datasets (millions→billions of pairs), report the Datacomp average plus subset scores, include ablations of hyperparameters (e.g., restart frequency and update count), and list batch sizes, samples processed, and optimizer choices.


3. A profiler table reports the NPN adds about 6–9% per-iteration time across representative encoders, do

**Weaknesses:**

1. The paper notes dataset download discrepancies vs. AmorLIP (different numbers of successfully fetched samples) yet still compares scores; large-scale web datasets are sensitive to crawl state, which can materially shift results. The largest-scale runs report single numbers without variability, and some baselines rely on third-party code with possible configuration drift—together reducing the strength of “method X > method Y” claims.


2. The estimation-error plots rely on a “true normalizer”; for global losses that sum over all negatives, computing (or approximating) this quantity exactly is costly and may itself introduce bias. The procedure for obtaining this target at scale is not fully detailed, which affects how to interpret the error curves.


3. While block-coordinate convergence is cited for idealized alternating optimization, the implemented procedure adds multiple NPN steps per batch and periodic re-initializations. There is no formal convergence rate or stability guarantee for this specific schedule, making the method’s robustness to optimizer/hyperparameter choices theoretically unclear.


4. Performance depends on the number of prototypes m, restart frequency, and update count; the paper shows drops when these deviate (e.g., too frequent restarts or too many NPN updates leading to batch-overfit), but provides limited guidance for transfer to different hardware/batch regimes. This can be a practical failure mode in low-resource or non-H100 settings.


5. All main experiments use 8×H100 GPUs and large global batches (e.g., 4096–5120). Claims about efficiency versus large-batch training would be stronger with results under constrained hardware and strict parity of training budgets and data filters across methods; otherwise, improved scores might partly reflect schedule choices rather than the optimization change alone.

**Questions:**

1. How is the “true” (log) normalizer computed for the error analyses at scale, and what bias/variance does that estimator have relative to the exact full-dataset partition values?
2. What are the peak memory and wall-clock contributions of the NPN as a function of prototype count m and embedding dimension, and how do these scale for larger backbones (e.g., ViT-L/H)? Please include per-step FLOPs and activation memory.
3. Does the alternating schedule with Tu>1 and periodic re-initialization admit any convergence guarantee (e.g., to a stationary point) under realistic stochasticity, or can it cycle?** Empirical stability diagnostics would help.
4. How sensitive are results to m and to the prototype initialization strategy? Would K-means/K-medoids over a streaming buffer of embeddings outperform random re-seeding?
5. Please report Datacomp and normalizer-error curves to validate the claimed robustness to small batches.
6. Does the NPN inadvertently learn to absorb temperature effects, and if so, how do you regularize to prevent degeneracy between τ and α(·)?
7. Can you provide ablations isolating the two accelerations (multiple NPN updates vs. periodic restart) across datasets, and quantify which contributes most under fixed compute?
8. Since dataset discrepancies materially affect AmorLIP/FastCLIP comparisons, will you release the exact URL lists and filtering scripts for each run so the community can reproduce your numbers byte-for-byte?
9. Does the NPN introduce a biased gradient for the encoder parameters relative to the true global objective? If unbiasedness does not hold, can you bound or characterize the induced bias and its dependence on m, Tr, and Tu?
10. How portable is the approach to multilingual CLIP, video-text, or retrieval-only training? Are any parts of the derivation tied to cosine similarity or image–text symmetry that would need adjustment?

**If you address my concerns, I will consider raising my score.**

---

> ### Author Response · Authors · 2025-11-21
> **Official Comment by Authors (Part 1 of 4)**
>
> We thank the reviewer for their detailed evaluation of our work.
>
> > **Q1**: The paper notes dataset download discrepancies vs. AmorLIP (different numbers of successfully fetched samples) yet still compares scores; large-scale web datasets are sensitive to crawl state, which can materially shift results.
>
> **A**: This appears to be a misunderstanding. AmorLIP was **re-run** and evaluated on the **same datasets** used for all other methods in our paper; therefore, the comparison with AmorLIP is fair. The note *“for AmorLIP, we observe variation between our results and their reported results, since the datasets we used are not exactly the same as theirs”* simply indicates that our versions of CC3M and CC12M  differ slightly from those used in their work, as they are web datasets and the downloaded snapshots vary across time.
>
> > **Q2**: The largest-scale runs report single numbers without variability, and some baselines rely on third-party code with possible configuration drift—together reducing the strength of “method X > method Y” claims.
>
> **A**: First, we note that, given the substantial computational cost of large-scale training, reporting a single result for large datasets is common practice in the literature—for example, in CLIP (Radford et al., 2021) and OpenCLIP (Cherti et al., 2023). Moreover, the improvements of our method are consistently observed across all five datasets. We will repeat the experiments three times on DFN-192M and DFN-1B and report the mean and standard deviation in the final version.
>
> Second, regarding the codebase, all methods are implemented using the same open_clip framework (Ilharco et al., 2021); hence **configuration drift does not exist**. OpenCLIP, SigLIP, and FastCLIP differ only in their loss function implementations. NeuCLIP extends FastCLIP by adding the NPN module, while AmorLIP extends OpenCLIP by incorporating its own loss and network design. To directly isolate and compare the key differences between NeuCLIP and AmorLIP, namely the objective function and the NPN architecture, we integrated AmorLIP’s design into our codebase and reported the results in Tables 14–16 of Appendix B.6. These findings confirm that the improvement of NeuCLIP is not due to configuration drift.
>
> > **Q3**:  How is the "true" (log) normalizer computed for the error analyses at scale, and what bias/variance does that estimator have relative to the exact full-dataset partition values?
>
> **A**: The "true normalizer" we computed is indeed the true normalizer without any bias. To get this quantity, we first obtain the embeddings $\mathbf{e}\_{1, i}, \mathbf{e}\_{2, i}$ for $\mathbf{x}\_{i}, \mathbf{z}\_{i}$ in the whole dataset using the model of our interest, which is done by performing forward pass on all the images and texts. Then for a given image $\mathbf{x}\_{i}$ and a given model, its true normalizer is computed using Equation (9). Similar procedure is applied for obtaining true normalizer for a given text. Thus the true normalizer does not incur bias or variance. To obtain the results in Figure 2\(c), we randomly sample **10K** data points, and for each data point, we compute its true normalizer and estimators from OpenCLIP, FastCLIP and NeuCLIP. Then we average the error over the 10K data points (and over 5 checkpoints) to get the results in Figure 2\(c).

---

> > ### Author Response · Authors · 2025-11-21
> > **Part 2 of 4**
> >
> > > **Q4**: About the theoretical convergence of NeuCLIP. Does the alternating schedule with $T_{u}> 1$ and periodic re-initialization admit any convergence guarantee (e.g., to a stationary point) under realistic stochasticity, or can it cycle?
> >
> > **A**: Thank you for the great question! Indeed, the convergence of block coordinate stochastic gradient methods with alternating single update on all blocks for general non-convex optimization problems has been established in [1].
> >
> > For the alternating schedule with **multiple updates** of the second block of variables (the NPN), under certain conditions (e.g., a Polyak-Lojasiewicz condition on the second block of variables, which has been proved for deep neural networks [2]), we can also establish convergence to a stationary point with respect to the first block of variables. The proof is similar to existing results for non-convex min--max optimization, e.g., [3]. The key idea is to show that $\|\|W\_{1} - W\_{1}\^{\*}(\mathbf{w},\tau)\|\|\_{2}\^{2}$ and $\|\|W\_{2} - W\_{2}\^{\*}(\mathbf{w},\tau)\|\|\_{2}\^{2}$ become small after a sufficient number of inner iterations, where $W_{1}^{\*}(\mathbf{w},\tau)$ and $W_{2}^{\*}(\mathbf{w},\tau)$ denote the closest optimal solutions of $W_{1}$ and $W_{2}$ given $\mathbf{w}$ and $\tau$ respectively. Once these error terms are controlled, the bias of stochastic gradient estimator for the first-block variables $(\mathbf{w}, \tau)$ can also be bounded to be small, ensuring convergence.
> >
> > *References*:
> > [1] Yangyang Xu, and Wotao Yin. "Block stochastic gradient iteration for convex and nonconvex optimization." SIAM Journal on Optimization 25, no. 3 (2015): 1686-1716.
> > [2] Zeyuan Allen-Zhu, Yuanzhi Li, and Zhao Song. "A convergence theory for deep learning via over-parameterization." In International conference on machine learning, pp. 242-252. PMLR, 2019.
> > [3] Zhishuai Guo, Yi Xu, Wotao Yin, Rong Jin, and Tianbao Yang. "Unified convergence analysis for adaptive optimization with moving average estimator." Machine Learning 114, no. 4 (2025): 1-51.
> >
> > > **Q5**: All main experiments use 8×H100 GPUs and large global batches (e.g., 4096–5120). Claims about efficiency versus large-batch training would be stronger with results under constrained hardware and strict parity of training budgets and data filters across methods; otherwise, improved scores might partly reflect schedule choices rather than the optimization change alone.
> >
> > **A**: Please note that on CC3M and CC12M we use just 1K and 2K batch sizes, respectively (cf. Table 1). Even the batch size of 4-5K is much smaller than that used in OpenAI's CLIP training and OpenCLIP training (32K to 88K). For all methods, we use exactly the same data, thus the parity of data filters is also enforced.
> >
> > We added the results of NeuCLIP with the same amount of compute as the baselines in Table 13 in Appendix B.6 in the revision, which we also present in the following table for your convenience. From the results we can observe that the performance of NeuCLIP slightly decreases but still outperforms others.
> >
> > | Method | CC3M | CC12M | DFN-14M | DFN-192M | DFN-1B |
> > | -- | -- | -- | -- | -- | -- |
> > | OpenCLIP | 21.84 | 27.91 | 37.78 | 54.58 | 56.25 |
> > | FastCLIP | 24.74 | 31.50 | 38.45 | 54.72 | 56.68 |
> > | SigLIP | 22.19 | 28.60 | 37.23 | 54.26 | 56.32 |
> > | AmorLIP | 22.89 | 29.86 | 37.53 | 53.83 | 56.24 |
> > | NeuCLIP | **25.06** | **31.75** | **39.16** | **54.85** | **57.28** |
> >
> > > **Q6**: Please report Datacomp and normalizer-error curves to validate the claimed robustness to small batches.
> >
> > **A**:  In the following table, we added additional experiments comparing the Datacomp Average performance of different methods using even smaller batch size (512) and constrained hardware (single GPU) with the same amount of compute on DFN-14M. We can see that our conclusions still hold.
> >
> > | | OpenCLIP | FastCLIP | SigLIP | AmorLIP | NeuCLIP |
> > | -- |  -- | -- | -- | -- | -- |
> > | Datacomp Average | 34.51 | 35.58 | 35.33 | 35.27 | **36.40** |
> >
> > The estimation error of different methods with batch size 512 on DFN-14M is presented in the following table, from which we can observe that NeuCLIP still achieves low estimation error. We also plot the curves and the results are presented in Figure 4(b) in Appendix B.6.
> >
> > | | OpenCLIP | FastCLIP | NeuCLIP |
> > | -- | -- | -- | -- |
> > | Estimation Error | 51.02 | 21.04 | 10.50 |

---

> > > ### Author Response · Authors · 2025-11-21
> > > **Part 3 of 4**
> > >
> > > > **Q7**: What are the peak memory and wall-clock contributions of the NPN as a function of prototype count $m$ and embedding dimension $d$, and how do these scale for larger backbones (e.g., ViT-L/H)? Please include per-step FLOPs and activation memory.
> > >
> > > **A**:
> > > 1. The overhead of maintaining and updating the NPN is $\mathcal{O}(dm)$, which is **minimal** compared to that of the backbone encoder. We report the peak memory overhead of different $m$ and $d$ on DFN-14M in the following table, where the percentage is calculated by dividing the additional memory of NeuCLIP by the memory of OpenCLIP.  For absolute numbers, please refer to  Table 6 in Appendix B.5.  We test $d$ in $[512, 768, 1024]$ since these are used by most CLIP models (including largest ones such as ViT-H/g). From the results we can observe that the memory overhead remains low.
> > >
> > >     | $d$ | $m=$ 1024 | 2048 | 4096 |
> > >     | -- | -- | -- | -- |
> > >     | 512 | 0.34% | 0.42% | 0.83% |
> > >     | 768 | 0.37% | 0.47% | 1.02% |
> > >     | 1024 | 0.46% | 0.64% | 1.30% |
> > >
> > >     In the following table, we present the per-iteration running time overhead of different $m$ and $d$ on DFN-14M, where the percentage is calculated by dividing the additional costs of updating NPN by the total costs of updating both NPN and the encoders. For absolute numbers, please refer to  Table 5 in Appendix B.4. From the results we can observe that the running-time overhead remains low.
> > >
> > >     | $d$ | $m=$ 1024 | 2048 | 4096 |
> > >     | -- | -- | -- | -- |
> > >     | 512 | 5.53% | 5.82% | 6.03% |
> > >     | 768 | 5.93% | 6.68% | 6.96% |
> > >     | 1024 | 6.35% | 6.81% | 7.45% |
> > >
> > > 2. When the backbones are scaled up, the memory consumption and running time of the CLIP model will increase, but that of the NPN remains the same as long as $m, d$ do not change. This means that the overhead of the NPN will decrease for larger backbones.
> > > 3. We measured the FLOPs and activation memory using `pytorch.utils.flop_counter` and `torch.profiler` on DFN-14M with batch size 512 per GPU (8 GPUs in total), and the FLOPs is 4.29 GFLOPs when $d=512$ and $m=4096$. As a reference, the amount of compute for CLIP ViT-B/32 alone is 7444.12 GFLOPs. For the activations, the value is 114.16MB with $d=512$ and $m=4096$, which is small when compared with other components.
> > >
> > > > **Q8**: How sensitive are results to $m$ and to the prototype initialization strategy? Would K-means/K-medoids over a streaming buffer of embeddings outperform random re-seeding?
> > >
> > > **A**:
> > > 1. We  added the ablation of the number of prototypes $m$ on DFN-14M and CC12M, and the Datacomp Average performance is presented in the following table. We observe that increasing $m$ helps improve the performance, but the results for all different $m$ are better than baselines.
> > >
> > >     | Dataset | $m=$ 1024 | 2048 | 4096 | 8192 |
> > >     | -- | -- | -- | -- | -- |
> > >     | DFN-14M | 38.57 | 38.56 | 39.16 | 39.25 |
> > >     | CC12M | 31.62 | 31.67 | 31.89 | 32.12 |
> > >
> > > 2. Also, we added results of K-means initialization over a streaming buffer on DFN-14M. Specifically, we maintain two buffers of size 64K to store image and text embeddings respectively. At each iteration, the buffers are updated in a first-in-first-out manner, where the latest features are added to buffer to replace the oldest features. When initializing the NPNs, we run K-means on all features in the buffers to select $m$ prototypes and assign them to the parameters of the NPNs. We conduct experiments comparing the K-means strategy with current strategy (random feature initialization), and present the results in the following table, from which we can observe that K-means leads to slight drop in performance. This is not surprising since the buffer consists of features from different iterations, which follow different distributions (since the model changes every iteration). And such distribution drift would hurt the effectiveness of K-means.
> > >
> > >     | Initialization | Datacomp Average | ImageNet & Variants | Retrieval |
> > >     | -- | -- | -- | -- |
> > >     | Features | 39.16 | 33.79 | 26.60 |
> > >     | K-means | 38.56 | 33.57 | 26.05 |
> > >
> > > > **Q9**: Does the NPN inadvertently learn to absorb temperature effects, and if so, how do you regularize to prevent degeneracy between $\tau$ and $\alpha(\cdot)$?
> > >
> > > **A**: No! The temperature parameter $\tau$ is still a variable to be learned in our method. In our experiments, we observed that the temperature $\tau$ converges to a similar value 0.01 as FastCLIP.

---

> > > > ### Author Response · Authors · 2025-11-21
> > > > **Part 4 of 4**
> > > >
> > > > > **Q10**: Can you provide ablations isolating the two accelerations (multiple NPN updates vs. periodic restart) across datasets, and quantify which contributes most under fixed compute?
> > > >
> > > > **A**: We added ablation study comparing three settings: (1) NeuCLIP (number of updates $T\_{u}= 10$, restart frequency $T\_{r}= 500$), (2) W/O Restart ($T\_{u}= 10$, $T\_{r}= \infty$), and (3) W/O Multiple Updates ($T\_{u}= 1$, $T\_{r}= 500$). We reduce the number of iterations for settings where $T\_{u}= 10$ so that the amount of compute across settings remains the same. We present the Datacomp Average performance on different datasets in the following table, from which we can observe that periodic restart contributes most to the performance.
> > > >
> > > > | Dataset | W/O Restart | W/O Multiple Updates | NeuCLIP |
> > > > | -- | -- | -- | -- |
> > > > | CC3M | 24.51 | 24.64 | 24.91 |
> > > > | CC12M | 31.22 | 31.48 | 31.72 |
> > > > | DFN-14M | 38.44 | 39.02 | 39.05 |
> > > >
> > > > > **Q11**: Since dataset discrepancies materially affect AmorLIP/FastCLIP comparisons, will you release the exact URL lists and filtering scripts for each run so the community can reproduce your numbers byte-for-byte?
> > > >
> > > > **A**: This appears to be a misunderstanding. All methods are run on the same datasets.  AmorLIP was **re-run** and evaluated on the **same datasets** used for all other methods in our paper; therefore, there is no dataset discrepancy  issue. The note *"for AmorLIP, we observe variation between our results and their reported results, since the datasets we used are not exactly the same as theirs"* simply indicates that our versions of CC3M and CC12M  differ slightly from those used in their work, as they are web datasets and the downloaded snapshots vary across time. Thus although the exact URL lists are released already, the community is unable to get exactly the same datasets because some links will have become invalid. We believe that they can reproduce the relative comparison between methods on the same dataset.
> > > >
> > > > > **Q12**: Does the NPN introduce a biased gradient for the encoder parameters relative to the true global objective? If unbiasedness does not hold, can you bound or characterize the induced bias and its dependence on $m$, $T_{r}$, and $T_{u}$?
> > > >
> > > > **A**: If we denote the joint objective by $F(\mathbf w, \tau, W_1, W_2)$, then the stochastic gradients of each variable are unbiased. But if we consider the objective in terms of $\mathbf w, \tau$ by optimizing $W_1, W_2$, i.e., $F_1(\mathbf w, \tau)=\min_{W_1, W_2}F(\mathbf w, \tau, W_1, W_2)$, then the stochastic gradients of $\mathbf w, \tau$ are still biased. That is why we use multiple iterations $T_u$ to update $W_1, W_2$ to control the bias. Please see response to Q4.
> > > >
> > > > > **Q13**: How portable is the approach to multilingual CLIP, video-text, or retrieval-only training? Are any parts of the derivation tied to cosine similarity or image–text symmetry that would need adjustment?
> > > >
> > > > **A**: Our framework of using NPN for reformulation can be applied to any objective that involves the structure $\frac{1}{n}\sum_{i=1}^n\log(\frac{1}{n}\sum_{j=1}^n\exp(\ell_{ij}(\mathbf w)))$, where the inner summation is over a large number of data. Hence, it is applicable to  multilingual CLIP, video-text, or retrieval-only training that uses contrastive losses.  The reformulation and derivation are NOT tied to cosine similarity or image–text symmetry. The only place that is tied to the cosine similarity is the design of the NPN network in Equation (11), which utilizes the optimal solution to $\alpha_1, \alpha_2$ in Equation (9). If $\ell_{ij}(\mathbf w)$ does not have any particular structure, one can consider a general MLP for the design of NPN.

---

> > > > > ### Comment · Reviewer_fUVb · 2025-11-25
> > > > >
> > > > > Thank you for the your rebuttal. Your responses address many of my concerns; if the final version can further 1) clarify the unbiasedness of the *true normalizer* error and 2) provide stronger justification for the stability of the alternating optimization procedure, I would like to increase my score.

---

### Official Review · Reviewer_erJT · 2025-10-29

**Soundness:** 3
**Presentation:** 3
**Contribution:** 3
**Rating:** 6
**Confidence:** 4

**Summary:**

The paper tackles the difficulty of estimating the partition function (normalizer) in CLIP’s global contrastive loss without relying on very large batch sizes. The authors (1) use convex conjugacy to rewrite each per-sample log-normalizer as the solution of a simple one-dimensional minimization, and (2) replace the n per-sample variables by a compact “normalizer-prediction network” (NPN) learned jointly with the CLIP encoders through a single objective. They propose an alternating optimization routine with two accelerators: several NPN steps per encoder update and periodic re-initialization of NPN “prototypes.” On several CLIP training scales (millions to a billion samples), NeuCLIP reports better Datacomp averages than OpenCLIP, FastCLIP, SigLIP, and AmorLIP under fixed compute budgets.

**Strengths:**

1. The convex-variational reformulation removes the reciprocal-of-estimator bias from mini-batch CLIP and avoids per-sample moving averages in FastCLIP. The unified loss couples encoder and NPN training without needing a separate consistency target for the normalizer.

2. Alternating updates with multiple quick NPN steps and periodic NPN restarts is easy to implement and, per their appendix, gives better stability than simultaneous updates.

**Weaknesses:**

1 **Limited accounting of compute and wall-clock.** Results are reported “under the same budget” and with “8 × H100,” but the paper does not provide thorough wall-clock and energy numbers for NeuCLIP vs. strong baselines at equal accuracy.

2. **Breadth of baselines and settings.**  SigLIP is included, but the study would benefit from (i) larger-batch SigLIP/OpenCLIP points at matched compute, and (ii) comparisons under stronger data filtering or with modern data recipes, since normalizer accuracy can interact with data quality. The paper notes some dataset mismatch with AmorLIP but does not fully normalize the comparison.

**Questions:**

1. **Compute and speed**. At matched Datacomp accuracy, what is the wall-clock time and GPU-hours for NeuCLIP vs. FastCLIP and SigLIP?

2. **Scaling to 1B+.** DFN-1B shows smaller gains than DFN-192M under the stated budget. Is this due to fewer total seen samples (1.0B vs. 1.3B) or to NPN capacity limits?

---

> ### Author Response · Authors · 2025-11-21
>
> We thank the reviewer for all the questions and suggestions.
>
> > **Q1**: DFN-1B shows smaller gains than DFN-192M under the stated budget. Is this due to fewer total seen samples (1.0B vs. 1.3B) or to NPN capacity limits?
>
> **A**: Thank you for pointing this out! This is due to fewer total seen samples. We ran another experiment on DFN-1B with 3B samples seen, and the results (presented in the following table) show larger gains, which confirms the effectiveness of the NPN. We have included the new results in the revision.
>
> | Dataset | # Samples Seen | OpenCLIP | FastCLIP | SigLIP | AmorLIP | NeuCLIP |
> | --- | --- | --- | --- | --- | -- | -- |
> | DFN-192M | 1.3B | 54.58 | 54.72 | 54.26 | 53.83 | **54.90** |
> | DFN-1B | 1B | 53.20 | 53.57 | 53.22 | 53.08 | **53.74** |
> | DFN-1B (New) | 3B | 56.25 | 56.68 | 56.32 | 56.24 | **57.34** |
>
> > **Q2**: At matched Datacomp accuracy, what is the GPU-hours for NeuCLIP vs. FastCLIP and SigLIP?
>
> **A**: In the table below, we report the percentage of GPU hours of NeuCLIP relative to FastCLIP or SigLIP when it achieves the same Datacomp Average performance as FastCLIP or SigLIP, with the computational overhead of NPN taken into account. The results show that NeuCLIP is faster than both FastCLIP and SigLIP for reaching the same performance level.
>
> | Methods | CC3M | CC12M | DFN-14M | DFN-192M | DFN-1B |
> | -- | -- | -- | -- | -- | -- |
> | NeuCLIP vs. SigLIP | 77.15% | 70.69% | 84.46% | 83.97% | 83.02% |
> | NeuCLIP vs. FastCLIP | 94.12% | 92.78% | 95.06% | 93.75% | 91.85% |
>
> > **Q3**: The study would benefit from (i) larger-batch SigLIP/OpenCLIP points at matched compute, and (ii) comparisons under stronger data filtering or with modern data recipes, since normalizer accuracy can interact with data quality.
>
> **A**:  First, the goal of this paper is to improve the training of CLIP models in resource-limited environments (e.g., 8 H100 GPUs).  Hence, comparing with larger-batch SigLIP/OpenCLIP is not our focus. Second, our datasets range from lightly-filtered data (CC3M, CC12M) to data filtered using the DFN pipeline.  We are not entirely sure what you mean by “modern data recipes.”  To the best of our knowledge, DFN represents a state-of-the-art data filtering methodology.
>
> > **Q4**: The paper notes some dataset mismatch with AmorLIP but does not fully normalize the comparison.
>
> **A**: This appears to be a misunderstanding. AmorLIP was **re-run** and evaluated on the **same datasets** used for all other methods in our paper; therefore, the comparison with AmorLIP is fair. The note *"for AmorLIP, we observe variation between our results and their reported results, since the datasets we used are not exactly the same as theirs”* simply indicates that our versions of CC3M and CC12M  differ slightly from those used in their work, as they are web datasets and the downloaded snapshots vary across time. We have clarified this in the revision.

---

### Official Review · Reviewer_LKoG · 2025-11-04

**Soundness:** 3
**Presentation:** 3
**Contribution:** 3
**Rating:** 6
**Confidence:** 3

**Summary:**

- Proposes NeuCLIP, a framework for efficient CLIP training by learning a neural normalizer that predicts log-normalization terms in contrastive loss.

- Reformulates the contrastive loss via convex and variational analysis, turning per-sample normalizer estimation into learning a compact neural network.

- Introduces an alternating optimization algorithm that jointly updates CLIP encoders and the normalizer-prediction network (NPN).

- Demonstrates consistent improvements over OpenCLIP, FastCLIP, SigLIP, and AmorLIP on datasets from 3M to 1B pairs.

**Strengths:**

- The paper is based on an elegant theoretical foundation combining convex conjugate and variational principles to remove per-sample normalizer tracking.

- We see strong empirical validation - NeuCLIP consistently outperforms baselines on multiple datasets and scales favorably to billion-sample training.

**Weaknesses:**

- Some hyperparameters (restart frequency, number of prototypes m) seem tuned per dataset; robustness to such choices is not discussed.

- It might be worth discussing the computational overhead of the extra NPN updates, restarts, and parameter sync cost to give practitioners more insights to use in reality.

**Questions:**

- What is the per-step wall-clock and memory overhead compared to FastCLIP and OpenCLIP?

- How sensitive are results to NPN hyperparameters?

---

> ### Author Response · Authors · 2025-11-21
>
> We thank the reviewer for their valuable comments and suggestions.
>
> > **Q1**: Some hyperparameters (restart frequency, number of prototypes $m$) are tuned per dataset. What is the robustness of these hyperparameters across different datasets? How sensitive are the results to NPN hyperparameters?
>
> **A**: We would like to note that we did not tune these hyperparameters for each dataset. Instead, we use the same restart frequency $T_{r}=500$, number of prototypes $m=4096$ across all datasets (cf. Table 4 in Appendix B.1).
>
> We presented the ablation study on different restart frequency $T_{r}$ in Figure 2(b) and Figure 5 on two datasets, DFN-14M and CC12M, respectively. We can see that $T_{r}=500$ yields the best performance on both datasets. On each dataset, varying the restart frequency could slightly affect the performance, e.g.,  39.16 ($T_{r}=500$) vs 38.41 ($T_{r}=20$) on DFN-14M.
>
> | Dataset | $T_{r}=$ 20 | 100 | 500 | 2500 | $\infty$ |
> | -- | -- | -- | -- | -- | -- |
> | DFN-14M | 38.41 | 38.54 | 39.16 | 39.06 | 38.48 |
> | CC12M | 31.59 | 31.64 | 31.89 | 31.47 | 31.22 |
>
> We also added the ablation on the number of prototypes $m$ on DFN-14M and CC12M, and the performance are presented in the following table. While $m=8192$ gives the best performance, all results here for different $m$ are better than other baseline methods.
>
> | Dataset | $m=$ 1024 | 2048 | 4096 | 8192 |
> | -- | -- | -- | -- | -- |
> | DFN-14M | 38.57 | 38.56 | 39.16 | 39.25 |
> | CC12M | 31.62 | 31.67 | 31.89 | 32.12 |
>
> > **Q2**: What is the computational overhead (wall-clock and memory) of NeuCLIP (the extra NPN updates, restarts, and parameter sync)?
>
> **A**: The overhead of maintaining and updating the NPN is **minimal** compared with that of the backbone encoder. The used NPN is small, whose number of parameters is **less than 8%** of the backbone’s parameters (please see the table below). The additional **per-iteration** training time introduced by NPN (including NPN updates, restart, parameter sync, etc) is already reported in Table 5 of Appendix B.4, and we present those results in the table below for your convenience, along with the number of parameters in encoders and in NPN.
>
> | Vision Encoder | Embedding Dimension| # Params of Encoder (M) | # Params of NPN (M) | Time of NPN (ms) | Total Time (ms) | Overhead |
> | -- | -- | -- | -- | -- | -- | -- |
> | ResNet50 | 1024 | 102.01 | 8.39 | 49.23 | 529.09 | 9.30% |
> | ViT-B/32 | 512 | 152.28 | 4.19 | 54.17 | 897.79 | 6.03% |
> | ViT-B/16 | 512 | 149.62 | 4.19 | 56.43 | 944.41 | 5.98% |
>
> We also report the additional memory consumption of NeuCLIP compared with OpenCLIP and FastCLIP, with results shown in the table below. As illustrated, NeuCLIP incurs only minimal memory overhead.
>
> | Vision Encoder | Embedding Dimension | NeuCLIP (MB) | OpenCLIP (MB) | Overhead over OpenCLIP | FastCLIP (MB) | Overhead over FastCLIP |
> | -- | -- | -- | -- | -- | -- | -- |
> | ResNet50 | 1024 | 10573.2 | 10340.1 | 2.25% | 10346.6 | 2.19% |
> | ViT-B/32 | 512 | 20721.0 | 20550.0 | 0.83% | 20583.2 | 0.67% |
> | ViT-B/16 | 512 | 55171.3 | 55006.5 | 0.30% | 55057.6 | 0.21% |

---

> > ### Comment · Reviewer_LKoG · 2025-11-27
> >
> > Thank you for the clarifications and new experiments, which addressed my questions. I will keep my score.

---

### Meta-Review · Area_Chair_5SFK · 2025-12-30

**Summary:**

The main weaknesses concern compute accounting, convergence and robustness guarantees, hyperparameter guidance, and some clarity in comparisons and related work. Reviewer LKoG initially worries that restart frequency and prototype count “seem tuned per dataset” and asks about per-step wall-clock and memory overhead. Reviewer erJT notes “limited accounting of compute and wall-clock,” suggests that efficiency claims would be stronger under constrained hardware and strictly matched budgets, and raises questions about smaller gains at DFN‑1B and interaction with data filtering and large batches. Reviewer fUVb flags potential dataset discrepancies versus AmorLIP, questions how “true normalizers” are computed, notes that the alternating schedule has no formal convergence rate, and sees sensitivity to m, restart frequency, and NPN updates as a practical risk, especially off H100-class hardware; they also ask about gradient bias and portability. Reviewer TYsU asks for more related work on embedding geometry, clearer retrieval evidence, and discussion of limitations.

The AC recommends acceptance, following reviewer majority opinion, because NeuCLIP offers a principled and well-executed framework for improving CLIP training under limited batch sizes, with sound convex–variational derivations, a simple but effective neural normalizer, and thorough large‑scale experiments showing consistent improvements over strong baselines, all with modest computational overhead.
On balance, AC agrees with positive points raised by all reviewers which outweigh the negative points. The authors are strongly encouraged to include the additional reviewer recommendations, experiments from rebuttal and clarifications in the camera-ready version.

**Reviewer Concerns:**

The authors’ rebuttal addresses many weaknesses. For hyperparams and overhead (LKoG), they clarify that the same restart frequency and prototype count are used across datasets and provide ablations showing modest sensitivity, along with tables quantifying per-iteration time and memory overhead (≈6–9% extra time, <8% of encoder parameters, and sub‑percent memory overhead across encoders). LKoG explicitly states that the “clarifications and new experiments” address their questions and keeps their score. For compute and fairness (erJT), they add matched‑compute comparisons where NeuCLIP still outperforms baselines, report GPU‑hour ratios showing NeuCLIP reaches the same Datacomp accuracy faster than FastCLIP and SigLIP, and provide small‑batch (512) and single‑GPU experiments demonstrating NeuCLIP’s advantage and lower normalizer error; they also clarify that all methods use the same data and OpenCLIP codebase, with AmorLIP re‑run on identical datasets. For convergence, unbiasedness, and robustness (fUVb), they explain that true normalizers are computed exactly by precomputing embeddings and summing over all negatives; they add a convergence argument in the appendix based on block‑coordinate SG and PL‑type conditions, and show that multiple NPN updates control bias; they present detailed scaling tables over m and embedding dimension, show that NPN overhead stays small and decreases relative to larger backbones, and add ablations on m and prototype initialization (including K‑means buffers), plus a study isolating the impact of multiple NPN steps versus periodic restarts (the latter contributing most). fUVb acknowledges that many concerns are addressed and indicates willingness to raise their score given the added unbiasedness and stability analysis. For related work and retrieval (TYsU), they incorporate EuCLIP/HyCoCLIP in the discussion as orthogonal directions, and present MSCOCO/Flickr recall results showing NeuCLIP’s retrieval gains; they also explicitly discuss NPN optimization as a limitation and venue for future work.

**Reviewer Scores:**

Post rebuttal, three reviewers (LKoG, erJT, TYsU) remain at 6 and explicitly confirm that they keep their scores, and the one initially at 4 (fUVb) states that their concerns are largely resolved conditional on clarifications that have been added in the revision. The residual issues—e.g., more exhaustive variability reporting on the largest datasets, stronger theory under fully realistic stochastic schedules, or more exploration under extreme resource constraints—are incremental rather than fundamental, and do not undermine the core contribution.

---

### Decision · Program_Chairs · 2026-01-26

Accept (Poster)